# Inhibition of epithelial cell YAP-TEAD/LOX signaling attenuates pulmonary fibrosis in preclinical models

Darcy Elizabeth Wagner [1,2,3,4,5,6,7,8] ✉, Hani N. Alsafadi [1,2,3,4,5],
Nilay Mitash [9,10], Aurelien Justet [11,12], Qianjiang Hu [9,10], Ricardo Pineda [9,10],
Claudia Staab-Weijnitz [13,14], Martina Korfei [15,16], Nika Gvazava [1,2,3,4],
Kristin Wannemo [9], Ugochi Onwuka [9], Molly Mozurak [9],
Adriana Estrada-Bernal [9], Juan Cala-Garcia [17,18], Katrin Mutze [5],
Rita Costa [5], Deniz Bölükbas [1,2,3,4], John Stegmayr [1,2,3,4],
Wioletta Skronska-Wasek [5], Stephan Klee [5], Chiharu Ota [5], Hoeke A. Baarsma [5],
Jingtao Wang [6,7], John Sembrat [9], Anne Hilgendorff [13], Jun Ding [6,7],
Andreas Günther [15,16], Rachel Chambers [19], Ivan Rosas [18], Stijn de Langhe [20],
Naftali Kaminski [11], Mareike Lehmann [5,13,21,22], Oliver Eickelberg [9] &
Melanie Königshoff [5,9,10] ✉

Idiopathic pulmonary fibrosis (IPF) is a progressive and lethal disease characterized by excessive extracellular matrix deposition. Current IPF therapies slow disease progression but do not stop or reverse it. The (myo)fibroblasts are thought to be the main cellular contributors to excessive extracellular matrix production in IPF. Here we show that fibrotic alveolar type II cells regulate production and crosslinking of extracellular matrix via the co-transcriptional activator YAP. YAP leads to increased expression of Lysl oxidase (LOX) and subsequent LOX-mediated crosslinking by fibrotic alveolar type II cells. Pharmacological YAP inhibition via verteporfin reverses fibrotic alveolar type II cell reprogramming and LOX expression in experimental lung fibrosis in vivo and in human fibrotic tissue ex vivo. We thus identify YAP-TEAD/LOX inhibition in alveolar type II cells as a promising potential therapy for IPF patients.

Idiopathic pulmonary fibrosis (IPF) is a fatal, chronic lung disease with unknown ethiology[1–5]. IPF is characterized by excessive extracellular matrix (ECM) deposition in the lung parenchyma resulting in progressive loss of lung function[6]. At present, there is no cure and median survival following diagnosis is between 2 and 3 years, worse than many malignancies. Recent studies have shown that distal lung epithelial injury and subsequent aberrant reprogramming of alveolar epithelial type 2 (AT2) cells may play a more prominent pathological role than previously realized, with several studies demonstrating anti-fibrotic effects when targeting fibrotic AT2 cells[7–11]. Moreover, several recent large-scale efforts to dissect the cellular landscape of IPF tissue using single cell RNA sequencing and spatial transcriptomics[12,13] have independently and collectively identified the appearance of diverse cell states in IPF, including populations of distal lung epithelial cells which are not detected in normal lungs[8,14–16].

The nuclear effectors of the Hippo pathway and known mechanotransducers, YAP and TAZ have recently been found to be expressed by transient alveolar epithelial cell types that are abnormally accumulated in IPF[17,18]. However, the mechanisms of how increased expression of YAP and/or TAZ in AT2 cells and their therapeutic

potential for human IPF, remains unexplored. Here, we explore the consequences of YAP activation in fibrotic AT2 cells. We demonstrate that fibrotic AT2 cells exhibit increased YAP activity and actively contribute to one of the main features of the fibrotic lung, increased ECM deposition. YAP regulates the expression and secretion of the ECM modifying enzyme lysyl oxidase (LOX) in fibrotic AT2 cells, which represents a previously unknown mechanism for AT2 cells to contribute to pulmonary fibrosis. Using the FDA-approved drug verteporfin (VP) and human fibrotic lung tissue cultures, we demonstrate that this mechanism is amenable to therapeutic intervention.

## Results

### Increased YAP activity in the alveolar epithelium in IPF

Recently published data from our group and others have identified increased YAP activity in the fibrotic alveolar epithelium[15,19–21]. Our single cell RNA sequencing (scRNASeq) dataset revealed cell-type specific gene expression of YAP, TAZ, and YAP target genes in alveolar epithelial cells with *YAP* as the highest expressed transcriptional coregulator of the Hippo pathway in alveolar epithelial cells, and increased expression of the YAP target gene connective tissue growth factor (CTGF) in transitional/aberrant alveolar epithelial cells in IPF, further indicating YAP activity is increased in IPF (Fig. 1A)[22,23]. We performed immunohistochemistry of control and IPF tissue and, confirming the scRNASeq data, found increased active, nuclear YAP expression predominantly in alveolar epithelial cells associated with regions of visible epithelial remodeling, while TAZ expression was found mainly in mesenchymal cells in fibrosis (Fig. 1B and Supplementary Fig. S1A) which matches other previous reports which focused on TAZ activity in fibroblasts[24,25]. Active YAP was found largely in aberrant distal lung epithelial cells marked by proSP-C positivity, a marker of AT2 cells, HOPX, marking transitional or AT1 cells, and KRT7, a marker of simple distal epithelial cells which is also highly elevated in aberrant basaloid populations. These cells were present in regions which were histologically abnormal (e.g. honeycombing regions), indicating either epithelial reprogramming of distal cells or migration of proximal cells into the distal region following injury (Fig. 1B and Supplementary Fig. S1A). Importantly, we also observed aberrant YAP activation in regions of IPF lungs with only moderate levels of histological injury, indicating that changes in YAP may be an early event in IPF development (Fig. 1B and Supplementary Fig. S1). In further support of this finding, we also found that nuclear YAP/TAZ is subjectively increased in the 'normal looking' regions of fibrotic murine lungs, including in alveolar type II cells marked by DC-Lamp (Fig. 1C). Importantly, these regions have not yet undergone extensive remodeling and still have reported stiffness values in the range of normal lung tissue[24,26]. In addition to mechanical stimuli such as stretch and stiffness, YAP is known to be regulated by the Hippo pathway, which is known to be active under conditions of high cell density regulating YAP cytosolic *versus* nuclear location. Thus, we examined Hippo signaling components (Supplementary Table S1) in the Lung Genome Research Consortium dataset, which contains whole lung homogenates of >100 patients with IPF. Notably, we found that Hippo pathway component gene expression was able to separate patients with IPF from 91 normal patients using PCA (Fig. 1D), however, no differences were observed in Hippo pathway components between normal patients and those with COPD ($n = 144$), indicating that loss of Hippo pathway components is not a universal feature of chronic lung diseases (Fig. 1D). Moreover, decreased expression of Hippo pathway components and increased expression of fibrotic markers (*MMP7* and *MUC16*) correlated with lung function (as assessed by %DLCO) and with patients with more severe disease (Fig. 1E). Together, these data support the hypothesis that YAP signaling is activated early in aberrant fibrotic AT2 in lung fibrosis, however, the functional impact of these findings and the potential consequences for therapeutic intervention, remains unknown.

### YAP-TEAD inhibition attenuates lung fibrosis in vivo

There are limited therapeutic options for IPF. Recent studies have started to investigate the involvement of YAP/TAZ in lung fibrosis in vivo. While a combined YAP/TAZ deletion in AT2 cells led to exaggerated lung fibrosis[27], we recently demonstrated that genetic loss-of-function of YAP (with preserved TAZ function) in AT2 cells led to attenuated pulmonary fibrosis[28]. In order to further gain insight into the in vivo function of YAP signaling, and to further extend studies to potential pharmacological treatment approaches for IPF, we next sought to determine whether pharmacological intervention with a known suppressor of the YAP-TEAD transcriptional complex and FDA approved compound, verteporfin (VP), could reduce fibrotic burden in vivo[29,30]. To this end, we used the well-established model of bleomycin-induced pulmonary fibrosis via intratracheal administration[31]. After lung injury was allowed to develop over the course of 7 days in vivo, a timepoint with well-established histological changes, we began administration of VP i.p. every other day until animals were euthanized on day 14 (Supplementary Fig. S2). We observed increased survival of animals which received bleomycin and were therapeutically treated with VP as compared to those animals who received vehicle only (survival: PBS-veh (10/10), PBS-VP (10/10), Bleo-veh (10/16), Bleo-VP (15/16)) (Fig. 2A, Supplementary Fig. S2B). Furthermore, we observed loss of nuclear Yap with VP treatment (Fig. 2B) and a significant decrease in *Ctgf*, a known downstream target of YAP-TEAD signaling (Fig. 2C). This was accompanied by a reduction of well-known fibrotic markers, including *Acta2*, *Serpine1*, and *Wisp1* (Fig. 2C). Animals treated with bleomycin and subjected to VP administration exhibited a reduction in cellular infiltrates and tissue density, as assessed histologically and quantified using the Modified Ashcroft Score (Fig. 2D, E). In support of these findings, total collagen content of the lungs of animals who received bleomycin and were treated with VP was significantly decreased as compared to animals who received bleomycin only (Fig. 2F). Interestingly, these animals exhibited loss of collagen despite the fact that VP treatment did not decrease the elevated transcriptional levels of major fibrillar lung collagens in bleomycin treated animals, *Col1a1* and *Col3a1* (Supplementary Fig. S3), indicating a potential post-translational mechanism of altered collagen deposition.

### YAP signaling induces LOX expression and secretion by AT2 cells

To further identify the potential mechanisms involved in YAP-driven profibrotic changes in the distal epithelium, we investigated primary normal and fibrotic AT2 cells isolated from murine lungs. Despite culture on stiff glass coverslips, nuclear expression of YAP/TAZ was significantly increased in fibrotic AT2 cells, while YAP/TAZ was found mainly in the cytosol of normal AT2 cells (Figs. 3A, B). In order to explore the role of YAP/TAZ in fibrotic alveolar cells, we silenced YAP/TAZ (siYT) in primary and normal fibrotic AT2 cells and performed transcriptome analysis (Supplementary Fig. S4A–G). In support of successful transcriptional knockdown, YAP/TAZ were among the most significantly downregulated genes in both normal and fibrotic AT2 cells, which we also confirmed on the gene and protein level along with reduction of *Ccn2(Ctgf)*, a well-known target of Yap/Taz (Supplementary Fig. S4A–C). Next, we explored differentially regulated genes following Yap/Taz knockdown which are elevated in fibrotic AT2 cells and reduced with siYT. We found that pathways associated with ECM and collagen assembly were among the most significantly enriched pathways, including collagen and lysl oxidase family members (Fig. 3D, Supplementary Table 2).

Fibrillar collagen assembly is regulated through a series of intracellular and extracellular processing steps, including collagen modifying and crosslinking enzymes. Notably, upon YAP/TAZ knockdown, only *Lox* was significantly reduced among the LOX family members in our transcriptome dataset (Fig. 3D, E). Therefore, we confirmed LOX

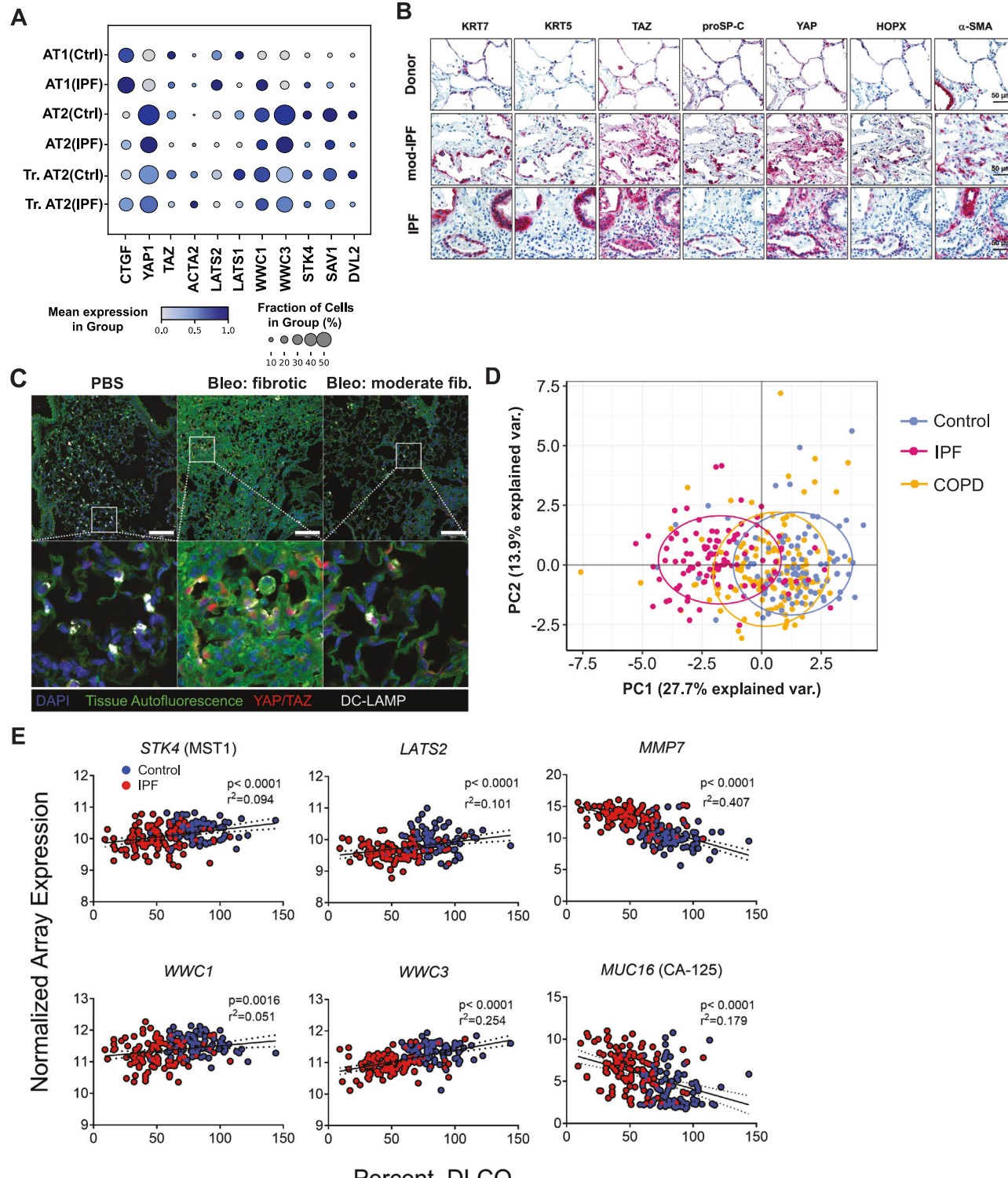

**Fig. 1 | YAP/TAZ is upregulated and active in the aberrant epithelium in IPF.**
**A** Publicly available scRNA-Seq data showing cell type specific gene expression in distal (alveolar) epithelial cells of IPF and Donor (Ctrl) lungs (IPFcellatlas.com: GSE135893). Changes in expression of Hippo family members, YAP, TAZ, and target gene CTGF in Donor *versus* IPF fibrotic epithelial cell subtypes is shown. (Tr. Transitional).
**B** Representative Immunohistochemical staining of YAP, TAZ, epithelial, and mesenchymal markers on tissue sections for Donor tissue, moderate-fibrotic IPF, and full fibrotic IPF tissues. $n = 6$ donor, $n = 6$ moderate-fibrotic IPF, $n = 13$ full fibrotic IPF.
**C** Representative immunofluorescence staining of Yap/Taz and Dc-Lamp, a marker of alveolar type II cells, on paraffin sections of murine lungs from PBS and bleomycin

treated mice after 14 days. Scalebar 20 um on the top row. Bottom row is a digital zoom 5x. $n = 5$ PBS, $n = 8$ Bleomycin. **D** Principal component analysis (PCA) of microarray analysis on samples from control subjects ($n = 91$), IPF ($n = 100$), and COPD ($n = 144$) patients from the Lung Genome Research Consortium (LGRC) dataset using Hippo pathway genes found in Supplementary Table 1. **E** Pearson correlation coefficient ($r^2$) between %DLCO measurements from Donor/IPF patients from the LGRC and gene expression of upstream Hippo signaling components *STK4, LATS2, WWC1, WWC3* and the profibrotic markers *MMP7* and *MUC16*. $p < 0.05$ considered significant; two-tailed. No adjustment made for multiple comparisons ($k = 6$ comparisons). Source data for Panels (**A, D**, and **E**) are provided as a Source Data file.

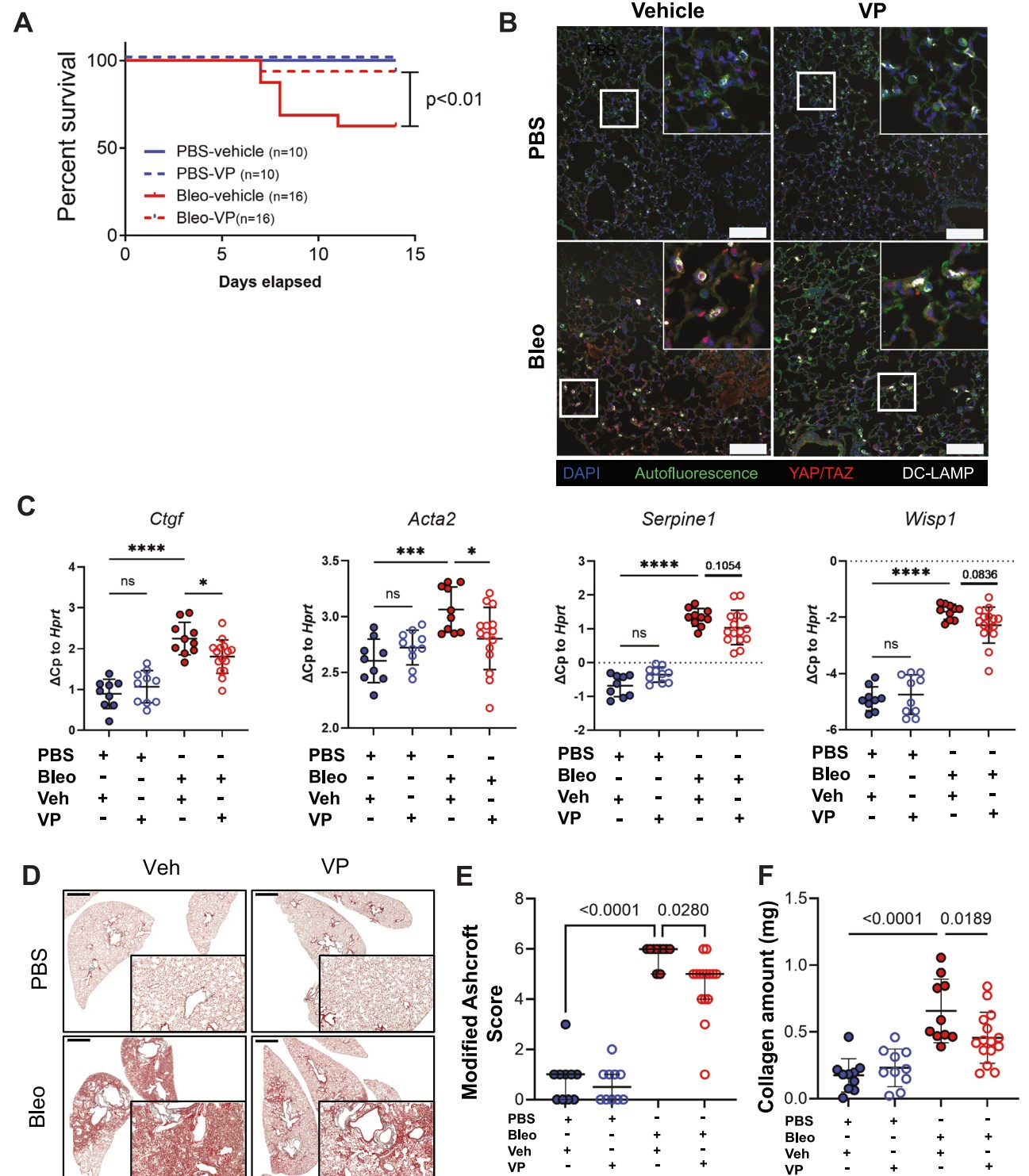

gene and protein expression in fibrotic AT2 cells and its reduction upon YAP/TAZ silencing, indicating that YAP/TAZ-driven LOX expression in fibrotic AT2 cells might contribute to the ECM niche in fibrosis (Fig. 3D, E). LOX is a well-known ECM modifying enzyme which has been previously shown to be elevated in murine models of lung fibrosis as well as in IPF patient samples[32]. It is critical for the extracellular assembly of the collagen triple-helix and deposition of both collagen and elastin in the lung. While LOX expression has previously been mainly associated with fibroblasts[33–36] which are well-known to be major contributors to ECM deposition and turnover, a role for distal epithelial derived LOX has not yet been described. Interestingly, LOX was the most highly enriched LOX-family member in diverse distal epithelial cells identified in a recently published spatial transcriptomics dataset (Fig. 3G) which minimizes cell biases which occur in single cell RNA seq technologies due to loss and/or preferential enrichment of certain cell types (e.g. over-representation of immune cells and underrepresentation of AT1 cells)[13]. In particular, alveolar cells, including aberrant basaloid cells had the most elevated expression of LOX in the spatial transcriptomic dataset. Despite differences in the cell types reported by the different technological platforms, this is in line with recent scRNASeq datasets of IPF lungs that have identified a subset of fibrotic epithelial cells with deranged ECM component

**Fig. 2 | Pharmacological inhibition of YAP-TEAD improves survival and attenuates experimental lung fibrosis in vivo. A** Survival analysis during 14 days of mice treated with PBS and Bleomycin (2U/kg) with or without verteporfin (VP) injections administered intraperitoneally every 48 h starting from day 7 to day 14 (45 mg/kg). *p < 0.05 considered significant: log-rank (Mantel-Cox) test. **B** Representative immunofluorescence staining of Yap/Taz in tissue sections from normal and fibrotic mouse lungs injected with verteporfin (VP) in vivo, n = 10 in each PBS group and n = 16 in each Bleo group from two independent sets of experiments. Scalebar 100 μm (Inlay 3X). **C** Gene expression of *Ctgf, Acta2, Serpine1*, and *Wisp1* relative to *Hprt* of tissue homogenates from study described in 2a. Each dot represents an individual mouse who survived until day 14 of the study. Mean ± s.d. *p < 0.05 assessed by one-way ANOVA with Holm-Šídák's multiple comparisons test. *Ctgf* (PBS-veh vs. PBS-VP, p = 0.3422; Bleo-veh vs. Bleo-VP, p = 0.0134; PBS-veh vs. Bleo-veh, p = 0.0002); *Acta2* (PBS-veh vs. PBS-VP, p = 0.2482; Bleo-veh vs. Bleo-VP, p = 0.0180; PBS-veh vs. Bleo-veh, p = <0.0001); *Serpine1* (PBS-veh vs. PBS-VP, p = 0.1054; Bleo-veh vs. Bleo-VP, p = 0.1054; PBS-veh vs. Bleo-veh, p = <0.0001); *Wisp1* (PBS-veh vs. PBS-VP, p = 0.5661; Bleo-veh vs. Bleo-VP, p = 0.0836; PBS-veh vs. Bleo-veh, p = <0.0001); **D** Representative Masson's trichrome staining of paraffin sections of PBS and Bleomycin mouse lungs treated with VP, scale bar: 2 mm. All animals were stained thus n = 10 for both PBS groups and n = 10 for Bleomycin and n = 15 for Bleomycin-VP. **E** Modified Ashcroft Score of the histological staining of the mouse lungs described in (**A**), each dot represents the median score of 3 independent experts who performed blinded analysis on digitized histological slides containing multiple sections per animal. Thus, each dot represents the histological score per animal. Due to ordinal data, median with 95% confidence interval shown. *q < 0.05 considered significant and assessed by Kruskal–Wallis, corrected for multiple comparisons using the two-stage linear step-up procedure of Benjamini, Krieger and Yekutieli (q value). PBS-veh vs. PBS-VP (q = 0.2969), PBS-Veh vs. Bleo-Veh (q = <0.0001), Bleo-Veh vs. Bleo-VP (q = 0.028). **F** Collagen amount as measured by high performance liquid chromatography of total tissue homogenates of mouse lungs described in (**A**); Each dot represents an individual mouse who survived until day 14 of the study. Mean ± s.d. *p < 0.05 assessed by one-way ANOVA with Holm-Šídák's multiple comparisons test. PBS-veh vs. PBS-VP (p = 0.4841), PBS-Veh vs. Bleo-Veh (p = <0.0001), Bleo-Veh vs. Bleo-VP (p = 0.0189) Source data for Panels (**A, C, E** and **F**) are provided as a Source Data file. Source Data for Panel (**D**) is deposited at S-BIAD1520.

expression[16]. We sought to confirm this finding on the protein level and therefore performed co-staining of LOX with YAP and HTII-280, a marker of alveolar type II cells in parallel sections of human control lung and IPF tissue sections. We found evidence of LOX expression in the lumens of honeycombing regions and LOX secretion adjacent to the apical surface of HTII-280+ cells with nuclear YAP. Taken together, these independent datasets confirm that LOX is elevated in AT2 cells in IPF (and/or aberrant epithelial cells which express HTII-280) with active YAP (Fig. 3G, H, Supplementary Fig. S5, S6).

### YAP-TEAD inhibition reduces collagen fibril formation by fibrotic AT2 cells

We next sought to determine whether the anti-fibrotic effects observed by VP in vivo were mediated by LOX. We found that LOX gene and protein expression was indeed reduced in whole lung homogenates from fibrotic mice that were treated with VP (Fig. 4A, B), while *Loxl1* was not (Supplementary Fig. S7). These results were further corroborated by a reduction of LOX expression in distal epithelium in situ (Fig. 4C). Treatment of AT2 cells with VP in vitro resulted in reduced Lox transcript and protein expression, further suggesting that AT2 cells are a major target for VP in vivo (Fig. 4D, E). Importantly, *Lox* knockdown in normal or fibrotic primary murine AT2 cells did not result in loss of the AT2 phenotype or induce senescence, indicating that targeting of epithelial *Lox* could be a promising strategy (Supplementary Fig. S8). Altogether, these data demonstrate that LOX is a downstream target of YAP-TEAD signaling in fibrotic AT2 cells and indicate that the fibrotic alveolar epithelium via active YAP-TEAD signaling and LOX secretion into the extracellular environment may contribute to extracellular collagen network formation in pulmonary fibrosis.

### Fibrotic AT2 cells modify the ECM niche by altering fibrillar collagen assembly

While AT2 cells are known to play a role in the formation of epithelial basement membrane formation mainly composed of laminins and network forming collagen IV, they have not been described to play a role in fibrillar collagen assembly (i.e. collagen I and III). In order to first determine whether secreted products from AT2 cells are sufficient to modify fibrillar collagen assembly, we established an assay whereby we incubated soluble telopeptide containing type I collagen in the presence of secretomes derived from normal and fibrotic AT2 cells, respectively (Fig. 5A). The telopeptide region of collagen at the N and C terminus has been previously shown to be the region in which LOX exerts its activity to stabilize the collagen triple helix via crosslinks; collagen type I is the major fibrillar collagen in the lung and most solid organs and is capable of self-assembly at 37 °C, including in the absence of cells. Interestingly, collagen networks formed in the presence of fibrotic AT2 cell secretomes were strikingly different in morphology as compared to normal AT2 cells and, in accordance with our in vivo data, supernatants from VP-treated fibrotic AT2 cells did not induce changes in collagen network formation as compared to the changes induced by fibrotic AT2 cell supernatants (Fig. 5B and Supplementary Fig. S9). Similar results were obtained from the secretome from fibrotic AT2 cells treated with siYT (Supplementary Fig. S10). Notably, fibrotic broncheoalveolar lavage fluid (BALF), representative of the in vivo secretome, increased maximum collagen gelation significantly (Fig. 5C, D), and this effect was reduced upon treatment with beta-aminopropionitrile (BAPN), a known LOX inhibitor which has previously shown efficacy in vitro for lung fibroblast driven collagen assembly[37,38]. Reduced collagen crosslinking by LOX inhibition with BAPN was further confirmed by western blot under reducing conditions (i.e. to reduce disulfide bonds between proteins) with consistent loss of the β-band at 250 kDa, representing intermolecular crosslinking (i.e. collagen monomer dimerization). This mimics previous observations where BAPN was shown to exert its action through reduction of the β-band in tissue-engineered fibroblast sheets to generate collagen networks with reduced crosslinking through LOX inhibition[38] (Fig. 5E). Finally, we sought to assess collagen crosslinks present in lung tissue of mice which had been treated with verteporfin in vivo using Fourier Transform Infrared Spectroscopy (FTIR) in ATR-Transmission mode. As expected, we observed significant upregulation of overall collagen content corresponding to mature collagen fibrils containing crosslinks at 1650 cm$^{-1}$ (Amide I) in PBS versus BLEO treated animals[39]. Interestingly, verteporfin caused a reduction in this peak as well as a decrease in the Amide II band at 1550 cm$^{-1}$. These data strengthen a putative role for active YAP-TEAD/LOX in the fibrotic alveolar epithelium driving altered extracellular collagen network crosslinking and thus formation in pulmonary fibrosis (Fig. 5F–H, Supplementary Table 3).

### YAP-TEAD inhibition reduces fibrosis in human pulmonary tissue ex vivo

Lastly, in order to explore whether targeting this identified YAP-TEAD/LOX axis is a potential therapy for patients with IPF, we examined the anti-fibrotic activity of VP in a highly complex ex vivo human 3D tissue culture model using precision cut-lung slices (PCLS). Previously, we have demonstrated that living human PCLS treated with a pro-fibrotic cocktail (TGFβ1, TNFα, PDGF-AB, and LPA) led to a robust fibrotic signature[40–43]. This human ex vivo model for IPF can successfully mimic injury and early fibrosis-like responses in human lung tissue. As our initial data indicates that modification of the ECM niche via YAP-TEAD activation in distal epithelium may be an early driver of disease (Fig. 1B and Supplementary Fig. S1), this model provides a unique opportunity to explore whether YAP-TEAD inhibition targets LOX expression and

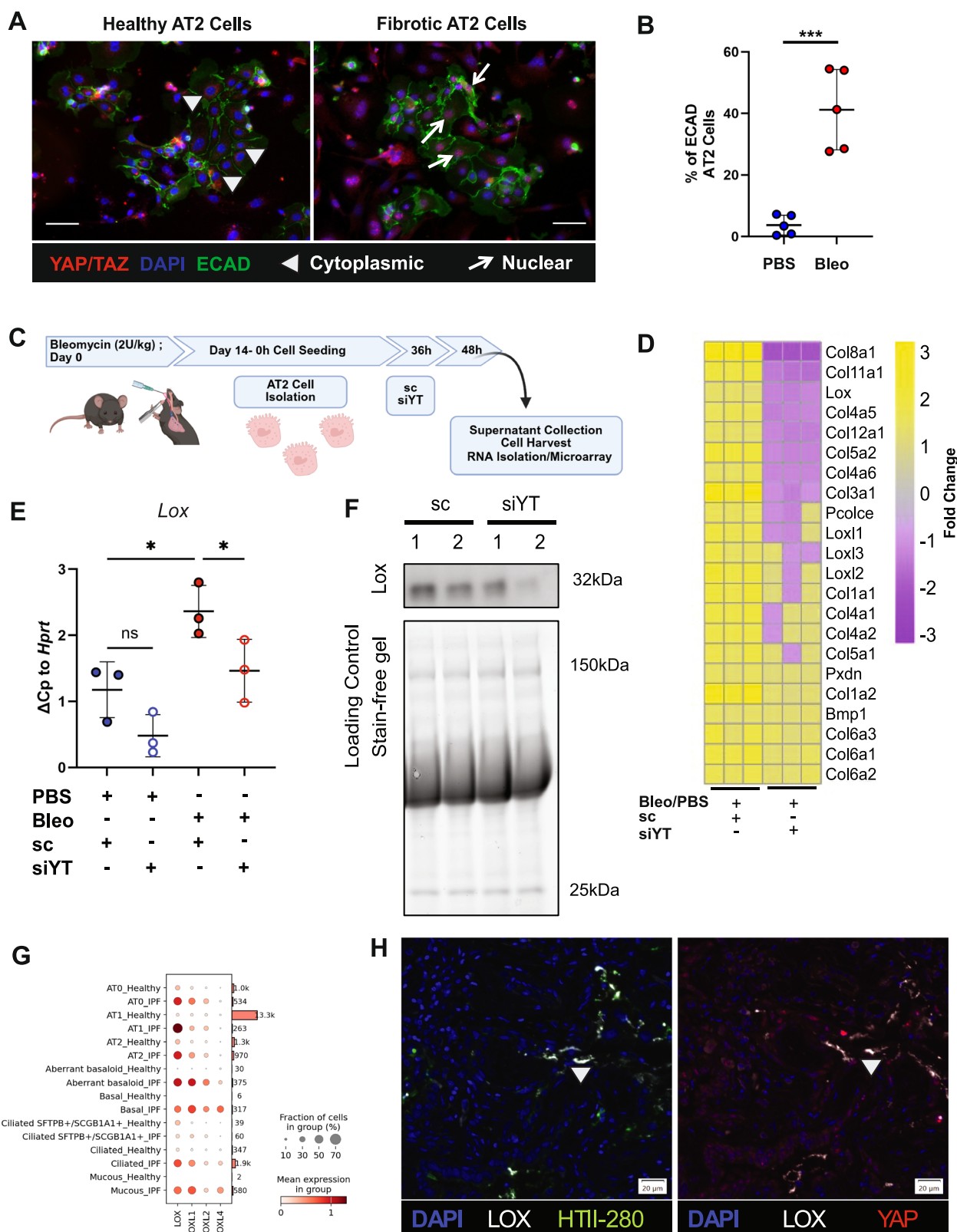

fibrosis markers in human lung tissue ex vivo. To this end, we subjected hPCLS to single nuclei RNA Sequencing (snRNA-Seq) analysis to further gain insights into cell-specific effects of VP. We profiled 76259 nuclei from a total of 24 PCLS obtained from two independent donor lungs each subjected to Fibrotic Cocktail (FC) with/without VP or the respective DMSO control, as previously described[40–43]. We identified all major expected cell lineages (epithelial, stromal, endothelial and immune) in our dataset (Supplementary Fig. S11) and found that FC increased known YAP-TEAD target genes *CCN1* and *CCN2* in epithelial cells (and *CCN2* to a lesser extent in fibroblasts), which were both downregulated with VP treatment (Supplementary Figs. S11, 12). We mainly focused on fibrotic epithelial cells for our further analysis due to our previous finding that YAP and LOX were elevated in the fibrotic distal epithelium. Five major epithelial cell types (Fig. 6A) based on

**Fig. 3 | YAP/TAZ in fibrotic AT2 cells regulate LOX secretion and collagen crosslinking. A** Representative immunofluorescence images (20× objective) of YAP/TAZ and E-cadherin in primary murine (pm) AT2 cells isolated from fibrotic and control murine lungs and plated on glass cover slips (200,000/cm²) (*n* = 5 each group). Scale bar = 50 µm. **B** Quantification of YAP/TAZ positive cells in healthy and fibrotic AT2 cells in (**A**). Quantification by image J of 5 independent pmAT2 isolates each, 5 fields per sample. Y-Axis depicts percent of ECAD positive AT2 cells with nuclear YAP/TAZ staining, Mean ± s.d. ***p* < 0.0001 unpaired, two-tailed *t*-test. *p* = 0.0003. **C** Schematic of experiments to silence Yap/Taz using siRNA (siYT) and the corresponding analyses. Created in BioRender. Wagner, D. (2025) https://BioRender.com/nmwzx11. **D** Heatmap of matrix genes from microarray data of pmAT2 cells isolated from bleomycin treated mice *vs* PBS treated mice followed by siRNA-mediated silencing of Yap/Taz (siYT), *n* = 3. Left 3 columns show fold change expression in fibrotic AT2 cells normalized with fold change expression in healthy AT2 cells. Right 3 columns show fold change expression in fibrotic AT2 cells subjected to siYT normalized with scramble control (sc). **E** Gene expression of lysl oxidase (*Lox*) in pmAT2 cells isolated from normal and fibrotic mice and culture with siYT, *n* = 3 independent isolations. Mean ± s.d. **p* < 0.05 paired one-way ANOVA with repeated measures and pre-selected comparisons. Corrected for multiple comparisons using Šidák's multiple comparisons test. PBS-sc vs. PBS-siYT (*p* = 0.1002); PBS-sc vs Bleo-sc (*p* = 0.0106); Bleo-sc vs. Bleo-siYT (0.0373). **F** Immunoblotting of Lox in the supernatants of study described in (**C**). **G** LOX-family members are elevated in epithelial cells in IPF tissue sections, with LOX prominently elevated in comparison to other LOX-family members (data extracted from ref. [13]). **H** YAP, LOX staining with HTII-280 (a marker of AT2 cells) in human lung sections of control and IPF tissue showing localization of LOX in areas of aberrant alveolar remodeling. Source data for Panels (**B**, **E** and **F**) are provided as a Source Data file. Source Data for Panel (**D**) is deposited E-MTAB-14643 and (**H**) is deposited at S-BIAD1520.

distinct markers with no discernible batch condition or cell cycle effects on data were identified (Fig. 6B, Supplementary Fig. S13). Similarly to a recent hPCLS study utilizing scRNA-Seq data[44], FC stimulation induced the emergence of aberrant basaloid epithelial cells (*KRT17* + *KRT5*−) at day 5 which we were able to further delineate into two separate subclusters which we termed aberrant basaloid-like 1 (ABL1) (elevated levels of *FN1*, *TNC*, *TGFBI*, *SERPINE1*) and 2 (ABL2) (elevated levels of *AREG* and *MUC21*) (Fig. 6A–C). Interestingly, ABL1 cells were enriched in genes associated with collagen matrix production (Fig. 6D) and while some ABL1 cells were present in the unstimulated conditions, the majority of these cells arose following FC stimulation indicating a population induced by FC rather than culture conditions (Fig. 6E). In addition, we identified a small cluster of basal cells (*TP63, KRT5, EYA2*) which only appeared under vehicle or FC + VP conditions but was noticeably absent from FC-only stimulated samples across patients (Fig. 6E). On the other hand, FC conditions induced the presence of a subcluster we termed inflammatory basal cells due to their co-expression of basal cell markers (*TP63*) and inflammatory markers previously associated with acute and chronically injured human airway epithelium (*ICAM1*[45]). Noticeably FC induced *LOX*, and to a lesser extent *LOXL1* and *LOXL2*, in the aberrant basaloid populations (Fig. 6F–H, Supplementary Fig. S14) and fibroblasts (Supplementary Fig. S15) which was subsequently decreased following VP stimulation (Fig. 6F, Supplementary Fig. S15). While VP stimulation, did not influence the epithelial cell type proportions (Fig. 6E), it significantly affected the gene signature of aberrant basaloid cells, including downregulation of *YAP* and *TAZ* (Fig. 6F, G), as well as significant decreases of ECM, profibrotic and EMT genes such as *TNC, SERPINE1, MMP9*, and *CXCL12* (Fig. 6G, Supplementary Fig. S16). These data support that notion that fibrotic epithelial cells, such as the aberrant basaloid cells present in IPF lungs, produce *LOX* and that VP treatment affects this cell type and induces anti-fibrotic effects.

The increase of LOX gene and protein expression upon fibrotic cocktail treatment and its reduction upon VP was further validated in an independent hPCLS cohort derived from 5 donor lungs upon fibrosis induction with FC (Fig. 7B–D, Supplementary Fig. S17). In addition, fibrotic marker gene expression was differentially reduced with VP treatment as compared to Pirfenidone and Nintedanib (Supplementary Fig. S18). Finally, we further validated our results in PCLS from end-stage IPF tissue explants and found that treatment with VP reduced *LOX* expression as well as *CTGF, WISP1* and *FN1* in IPF tissue (Fig. 7E, F). Interestingly, VP treatment led to an increase in epithelial/AT2 cells marker *ECAD* and *SFTPC*, suggesting a beneficial effect of VP on epithelial cell function in IPF.

Overall, we report a mechanism on how fibrotic AT2 cells modify the ECM niche, including fibrillar collagen assembly, which is driven through a deranged YAP/TEAD-LOX axis. By modifying the ECM niche, AT2 cells might play a major role in fostering a self-perpetuating vicious cycle of YAP-TEAD/LOX activation driving fibrosis (Fig. 8).

Inhibition of this axis by VP led to attenuation of fibrosis in vivo as well as in human pulmonary fibrosis ex vivo thus representing a potential therapeutic avenue.

## Discussion

Pulmonary fibrosis is a devastating progressive lung disease with limited therapeutic options. Here we identify aberrant YAP/TEAD/LOX signaling in fibrotic alveolar cells as a central mechanism in pulmonary fibrosis onset and progression. We demonstrate that fibrotic alveolar cells exhibit increased YAP activity and actively contribute to one of the main features of the fibrotic lung, increased ECM secretion, deposition and crosslinking. YAP regulates the expression and secretion of the ECM modifying enzyme lysyl oxidase (LOX) in fibrotic alveolar cells, which represents a previously unknown mechanism for AT2 cells to contribute to pulmonary fibrosis. Using the FDA-approved drug verteporfin and human fibrotic lung tissue cultures, we demonstrate that this mechanism is amenable to therapeutic intervention. We build upon and extent recent studies that highlight the diverse and dynamic roles of YAP/TAZ signaling in lung injury and (impaired) repair[17,19–21,24,26–28,46–49]. Early YAP/TAZ studies in the lung have largely focused and established its mechanotransduction role in pulmonary fibroblasts[24,47,50,51] however, the observation that YAP/TAZ are further activated in lung epithelial cells, has only emerged of late and the mechanistic and functional consequences of epithelial cell YAP/TAZ signaling represent an area of active investigations. A critical role for YAP/TAZ signaling in the lung epithelium in the context of pre- and postnatal lung development has been found, in which YAP has been shown to regulate cell differentiation including proximal airway differentiation[52], YAP and TAZ in AT2 to AT1 differentiation[46,48,49,53–55] as well as bronchial morphogenesis[56,57] and sacculation[58]. The functional contribution of YAP and/or TAZ in the context of lung fibrosis, which is characterized by dramatic epithelial cell reprogramming[59], is less explored. Gokey et al. have reported increased YAP activity in bronchial epithelial cells in fibrosis, leading to mTOR/PI3K/AKT-driven cell proliferation and migration, supporting a profibrotic role for YAP in the airways[20]. Similar findings have been reported by Stancil et al. in airway epithelial cells, identifying ERBB/YAP signaling as a potential driver of airway remodeling[60]. We recently showed that Yap but not Taz lead to bronchiolization in lung parenchyma through cooperation with Myc[61]. These initial observations are extended in our studies, presenting several lines of evidence that fibrotic AT2 cell states in the human and mouse lung exhibit high levels of nuclear YAP. We demonstrate that fibrotic YAP-active AT2 cells contribute to collagen crosslinking and ECM deposition. We identify a previously unknown profibrotic mechanism, in which YAP expression does not only contribute to changes in the cellular phenotype of fibrotic AT2 cells but further drives the expression of the ECM crosslinking enzyme LOX leading to increased ECM/collagen formation and crosslinking in pulmonary fibrosis. YAP1 has been found to be enriched in the LOX

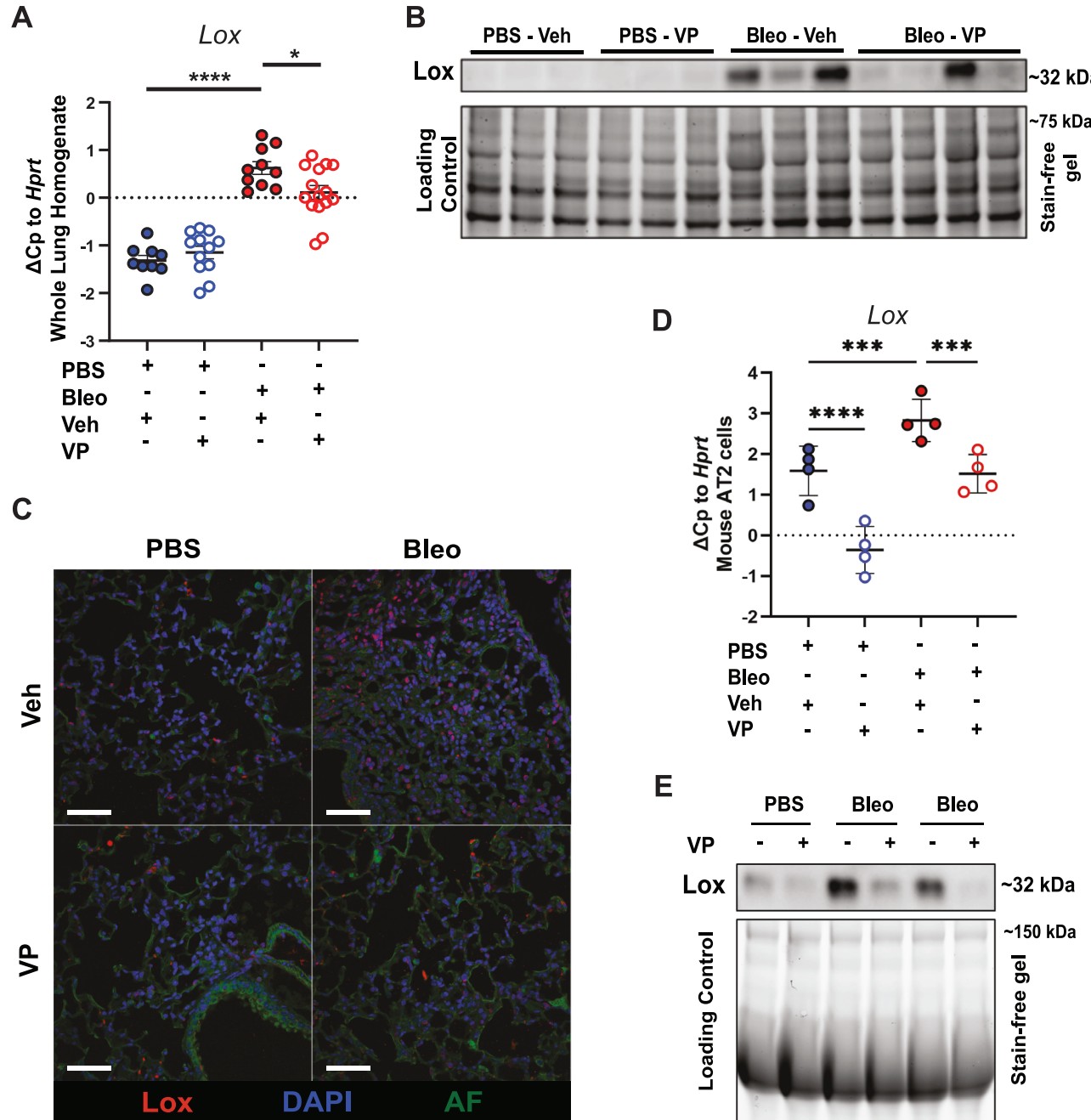

**Fig. 4 | YAP/TEAD inhibition by verteporfin reduces Lox expression and ECM crosslinking in fibrotic AT2 cells. A** *Lox* gene expression (*n* = 10 each) (*adjusted *p* < 0.05 considered significant with one-way ANOVA with Holm-Šídák's multiple comparisons test). Mean ± s.d. PBS-veh vs. PBS-VP (*p* = 0.5039), *p* < 0.0001 for PBS-veh vs Bleo-veh, Bleo-veh vs. Bleo-VP (*p* = 0.400). **B** Immunoblotting of Lox protein on tissue homogenates of normal and fibrotic mouse lungs injected with VP in vivo, *n* = 3–4 as shown. **C** Representative immunofluorescence staining of Lox in tissue sections obtained from the same study described in (**A**) and (**B**) (Supplementary Fig. S2), scale bar: 50 um. (AF Autofluorescence). **D** *Lox* gene expression. Mean ± s.d.

*adjusted *p* < 0.05 considered significant with one-way ANOVA with repeated measures and pre-selected comparisons. Corrected for multiple comparisons using Holm-Šídák's multiple comparisons test. PBS-veh vs. PBS-VP (*p* < 0.0001); PBS-veh vs Bleo-veh (*p* = 0.00009); Bleo-veh vs. Bleo-VP (0.0006). *n* = 4 independent isolations and **E** immunoblotting of Lox protein in supernatants of pmAT2 cells isolated from normal and fibrotic mice and treated with VP for 48 h, representative image, *n* = 4. Equal protein was loaded into each well and confirmed through the use of stain-free gels which label tryptophans. Source data for Panels (**A**, **B**, **D** and **E**) are provided as a Source Data file.

promotor region in a cancer cell line, suggesting a direct effect of YAP1 on LOX expression[62]. To this end, we demonstrated that YAP inhibition, using either a pharmacological compound (VP) or siRNA-mediated knockdown, reduces LOX expression in AT2 cells in vitro, experimental lung fibrosis in vivo, and human lung fibrosis modeled in PCLS ex vivo.

YAP activation in fibrosis has been largely investigated in the context of its role as a mechanotransducer in mesenchymal cells,

which is activated due to increased stiffness in fibrosis. However, recent work has also examined a role for activated YAP/TAZ in endothelial cells in the context of aging-associated fibrosis[63]. Furthermore, several recent publications have identified dysregulated Yap/Taz or TEAD signaling networks in epithelial cells through bioinformatic approaches[12,21] but did not follow up on these findings. We identified increased epithelial YAP/TAZ activity even in 'normal looking' regions of fibrotic lungs, in which little changes in stiffness have been

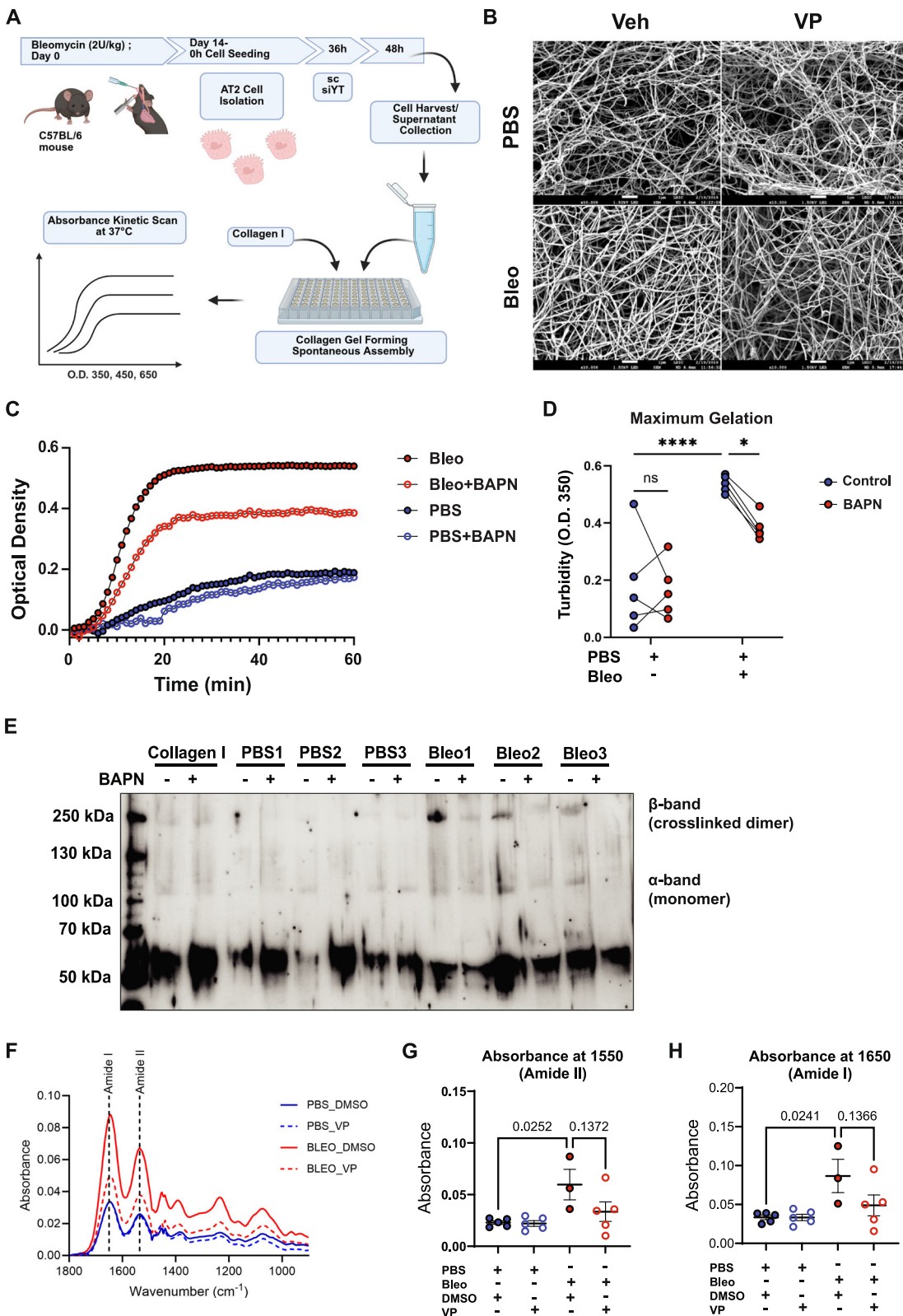

described[64], suggesting a previously unknown role for the activation of YAP that is independent of ECM stiffness. YAP is regulated by the developmental Hippo pathway[65] and we found that loss of Hippo pathway components in IPF correlated with YAP/TAZ activation, leading to the hypothesis that loss of this pathway in normal epithelium might be sufficient to drive distal epithelial cells towards aberrant cellular phenotypes observed in IPF. This is substantiated by previous studies, which have found dysregulated Hippo component transcripts in single cell RNAseq data sets[14,20]. Accumulating evidence in the adult lung suggests a role of the Hippo components and upstream YAP regulators serine/threonine-specific kinases (STKs) Mst1 and Mst2 as well as Lats 1 and 2 in lung development and adult regeneration. Loss of Mst1 and 2 revealed distorted lung maturation with decreased alveolar sacculation and increased cellularity, features also seen in pulmonary fibrosis[66]. We recently showed that loss of Mst1/2 or Nf2, another upstream Hippo kinase, in AT2 cells stabilizes Taz, permitting its

**Fig. 5 | AT2 cell derived LOX mediates collagen gelation. A** Schematics for collagen assembly assay with supernatants from pmAT2 cells isolated from normal and fibrotic mouse lungs were treated with siYT. Rat-tail type I collagen was used in a carbonate buffer at 1 mg/ml, pH 7.8, and 37 °C with the addition of biological sample, and turbidity is monitored with optical density measurement for the first two hours. Immunoblotting of collagen I and scanning electron microscopy (SEM) of the resulting gels were used as endpoint measurements, n = 3. Created in BioRender. Wagner, D. (2025) https://BioRender.com/nmwzx11. **B** SEM images of collagen formation assay described in Fig. 5A and performed on supernatants from pmAT2 cells isolated from normal and fibrotic mice and treated with VP for 48 h, n = 4 animals. scale bar: 1 um (×10,000). **C** Collagen gelation kinetic curves using BALF treated ex vivo with beta-aminopropionitrile (BAPN) from animals treated for 14 days with PBS or Bleomycin. Each line represents the average of n = 5 animals per condition. OD350nm. **D** Maximum gelation of curves in (**C**); Two-way ANOVA with uncorrected Fisher's LSD. PBS (Control vs. BAPN, p = 0.7197), Bleomycin (Control vs. BAPN, p = 0.0162), Control (PBS vs. Bleomycin, p < 0.0001); **E** Immunoblotting of collagen I of gels from (**D**) after gelation for 48 h at 37 °C. **F–H** FTIR spectra and peak quantification from lung tissue sections of one cohort from PBS or Bleomycin-treated animals who received VP or DMSO controls. N = 5 for PBS, n = 3 for BLEO-DMSO and n = 5 for BLEO-VP; **F** mean of all samples shown without error bars. **G**, **H** Mean ± s.d. p-values indicated for one-way ANOVA with Tukey posthoc test. p < 0.05 considered significant. Source data for Panels (**C–H**) are provided as a Source Data file. Source Data for Panel (**B**) is deposited at S-BIAD1520.

translocation into the nucleus to drive AT1 cell differentiation. Interestingly, loss of Taz, but not Yap, in AT2 cells inhibited differentiation to AT1 cells, indicating divergent roles of Yap and Taz in AT2 to AT1 differentiation[61]. Loss of Lats1/2 in airway secretory cells (i.e. Scgb1a1) has been shown to induce Yap/Taz activation which was accompanied by loss of the secretory phenotype, acquisition of transitional markers and evidence of subepithelial fibrosis, including activation of fibroblasts and collagen deposition[67]. Together, these studies point to cell and context dependent effects of Yap and Taz which are also regulated by loss of Hippo pathway components in the lung epithelium.

Increased ECM deposition is a consequence as well as a driver of pulmonary fibrosis, leading to a positive feedback loop[68,69]. Several data suggest that changes in the ECM alone are sufficient to induce alteration of normal cell types towards pro-fibrotic phenotypes[69,70]. The ECM in the IPF lung is highly dynamic and far from a static environment with little remodeling, as initially thought. Multiple ECM components (termed the matrisome) are continuously secreted, deposited and degraded in the IPF lung, with an overall sum of net deposition further remodeled and stiffened by crosslinking enzymes of e.g. lysyl oxidase (LOX) family[71]. ECM crosslinking is a key process that perpetuates the profibrotic niche in IPF. The LOX/LOXL proteins have been put forward as potential targets for the treatment of IPF[35]. Previous studies have shown that LOX and the four LOX-like proteins (LOXL1-4) play a key role in crosslinking of the ECM[33–37,72]. Inhibition of LOX using β-Aminoproprionitrile (BAPN) reduced experimental pulmonary fibrosis[37]. While LOX expression has previously been detected in fibroblasts, the role of LOX expression in fibrotic AT2 cells has not been investigated. Our observations are in line with Aumiller et al demonstrating that LOX family members are expressed by epithelial cells in pulmonary fibrosis[32], with the highest expression of LOX and LOXL4. Interestingly, in contrast to LOXL2, LOX expression was not induced by TGF-β1 in bronchial epithelial cells. LOX expression has been shown to be regulated by TGF-β1 in fibrosis in other cell types, such as fibroblasts[73]; our data indicate that LOX expression in distal epithelial cells might be driven by other upstream factors of YAP, such as the Hippo pathway. Interestingly, one of the original studies that identified TEADs as the main transcription factor family interacting with YAP identified LOX but not LOXL2 as a downstream target of YAP-TEAD interactions in breast epithelial cells[74]. A clinical trial targeting LOXL2 using the monoclonal antibody simtuzumab, found no benefit regarding survival[75], although limited data about its bioavailability and target engagement in IPF are known. In our current study, we targeted the distal YAP-TEAD LOX axis using multiple distinct approaches which together strongly supports a role for distal epithelial-derived LOX in altered collagen assembly in the parenchyma in pulmonary fibrosis. First, we utilized BAPN to inhibit LOX in a real-time collagen assembly assay utilizing BALF from a murine model of pulmonary fibrosis. BAPN inhibits intra- and intermolecular covalent cross-linking of collagen and elastin connective tissue proteins[76]. We found that LOX inhibition by BAPN reduced real-time collagen assembly induced by fibrotic BALF. While BAPN is described as a potent and irreversible LOX inhibitor, it has also affinity for other LOX family members and for other

amine oxidases[77]: thus, we cannot exclude off target effects as potential co-mediators of the effects we see due to the complex composition of BALF. However, we observed similar alterations in collagen assembly using supernatants from fibrotic alveolar cells with YAP/TAZ knockdown as well as those treated with verteporfin (VP). Therefore, our data strongly points to a mechanism by which altered YAP-TEAD activity induces LOX expression in distal epithelial cells which in turn alters local collagen assembly.

In order to show proof of principle for therapeutic inhibition of YAP-TEAD signaling via verteporfin, an FDA-approved drug for the treatment of macular degeneration, we utilized in vitro, in vivo and ex vivo assays utilizing human tissue. VP led to a reduction in LOX expression across all experimental conditions and furthermore reduced overall fibrosis severity in fibrotic AT2 cells in vitro, as well as in murine lungs in vivo and human fibrotic PCLS ex vivo. Importantly, VP treatment reduced the total amount of collagen, including the amount of crosslinked fibrillar collagen as assessed by FTIR. Thus, we identify a potential therapeutic approach, which targets cellular reprogramming of alveolar cells to aberrantly express LOX and regulate ECM deposition. Therapeutically targeting epithelial cells is an emerging approach in IPF[78], and our data further strengthen this rationale by providing a previously underrecognized contribution of these cells to ECM deposition and remodeling. All together, we hypothesize that temporal dysregulation of Hippo components leads to YAP activation in AT2 cells in early stages of IPF which initiates alveolar reprogramming and onset of remodeling, while stiffness is an additional contribution in later stages of fibrosis.

We used a pharmacological approach to therapeutically intervene with YAP-TEAD signaling and demonstrated that the FDA approved drug verteporfin exhibited inhibitory effect of pro-fibrotic epithelial cells functions. While using this approach corroborates the potential translatability of our study, pharmacological approaches also exhibit limitations. We cannot exclude that VP treatment in vivo and ex vivo also affects other cell types; in fact, VP treatment on YAP/TAZ activated fibroblasts may be beneficial and we did observe evidence of inhibition of pro-fibrotic markers in fibroblasts in our snRNAseq dataset. Future studies should examine this in more detail as well as characterize other ECM deposition versus degradation pathways. Nonetheless, our snRNA-seq data and in vitro gelation assays supports the fact that VP indeed targets epithelial cells. snRNA-Seq analysis of VP-treated human fibrotic tissue revealed a strong effect of VP on the phenotype of aberrant basaloid cells and their LOX expression, which represent the dominant fibrotic epithelial cells in the fibrotic lung[8,79]. Recent single cell data from the hPCLS model predicted that these cells are derived from AT2 cells[44]. While this in silico based hypothesis is further corroborated by recent organoid data from us[80] and others[81] that FC treatment or mesenchymally-derived factors induce differentiation of human AT2 cells or iAT2 cells into aberrant basaloid cells (KRT17+, KRT5−), respectively, lineage tracing studies are needed to further test these assumptions.

Moreover, we recently reported that genetic loss-of-function of YAP, but not TAZ, in AT2 cells (driven by SPC) led to attenuated pulmonary fibrosis in vivo[28], thus corroborating our findings. Of note, a

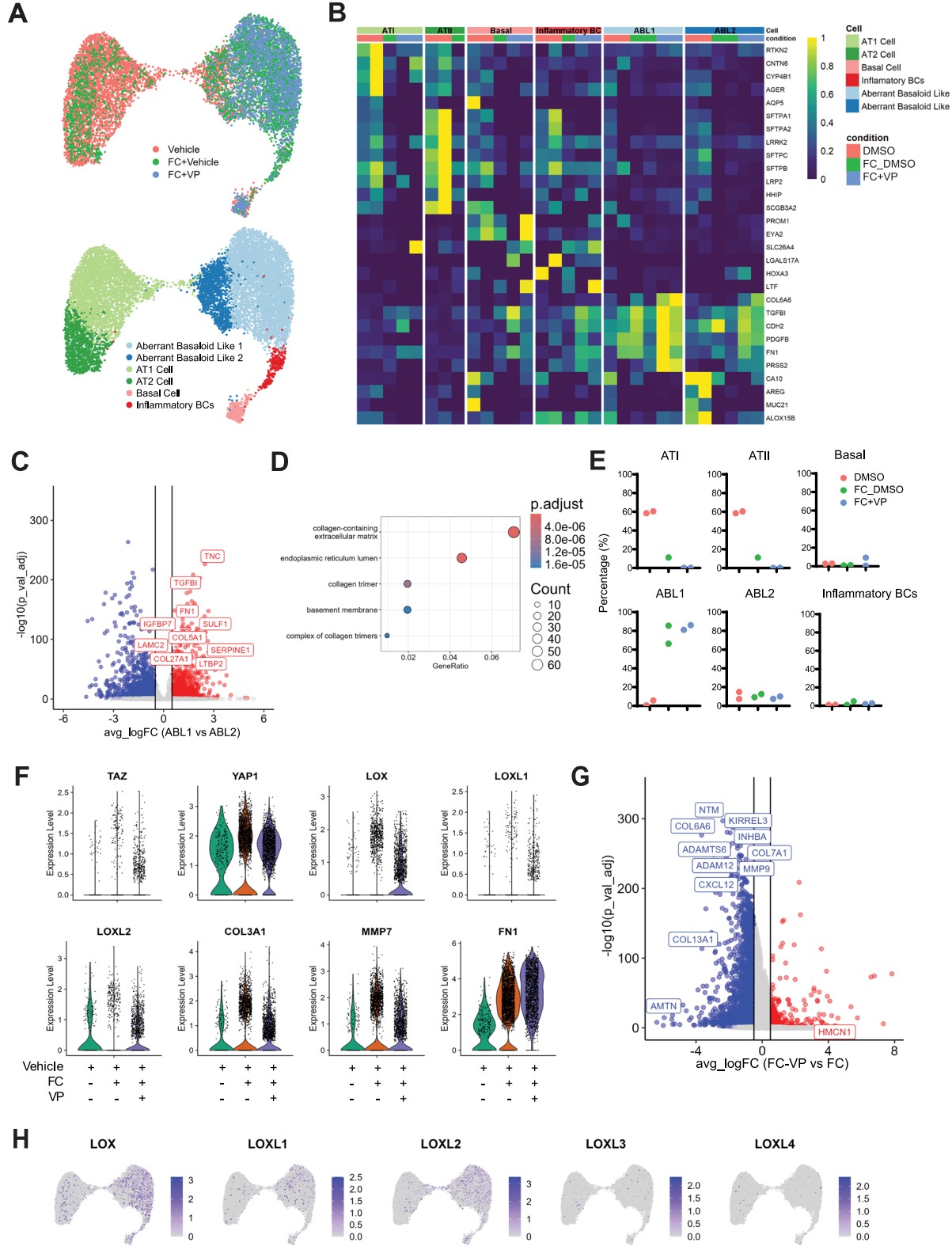

recent study using combined YAP/TAZ deletion in AT2 cells demonstrated increased inflammation and failed repair in two different models of alveolar injury[27]. Similarly, we recently showed that combined loss of Yap and Taz in bronchial epithelial cells (Sox2+) resulted in decreased AT1 cell regeneration and increased pulmonary fibrosis following bleomycin injury[61]. However, we found that Yap and Taz have divergent roles in epithelial repair. Active Taz in AT2 cells regulates AT1 cell differentiation while active Yap regulates basal-like cell mediated bronchiolization through cooperation with increased Myc levels[61]. Interestingly, YAP deletion activates TAZ signaling, which can contribute to repair in vivo. These studies highlight the importance of furthering our understanding of the separate roles of YAP *versus* TAZ signaling as most data in the literature is based on effects elicit by combined YAP and TAZ.

YAP and TAZ have independent roles as co-transcription factors and are known to interact with a variety of transcription factors[82]. It is

**Fig. 6 | YAP-TEAD inhibition by verteporfin reduces profibrotic genes in human fibrotic lung tissue ex vivo. A** Uniform Manifold Approximation and Projection (UMAP) representation of 11753 epithelial nuclei from 24 PCLS generated from 2 control donor lungs (4 PCLS per subject and per condition); the UMAPs are labeled by condition (4 PCLS per subject and per condition): Vehicle (Veh), Fibrotic Cocktail (FC)+Veh, FC+Verteporfin (VP), (top panel) or by cell type, Aberrant Basaloid Like 1 and 2; alveolar type 1 (AT1) cell; alveolar type 2 (AT2) cell, Basal cell; and Inflammatory Basal Cells (BCs)(bottom panel). **B** Heat map of normalized gene expression per cell type within each condition; Columns are ordered by disease status and cell type. **C** Differential gene expression between the two aberrant basaloid nuclei clusters, genes labeled in red are upregulated in aberrant basaloid like 1 cluster (log2FoldChange > 0.5; adjusted *p*-value < 0.05, Bonferroni correction in FindMarkers()); labeled genes are selected based on GO enrichment in (**D**).

**D** Gene Ontology enrichment based on the upregulated genes in aberrant basaloid Like 1 cells in comparison to aberrant basaloid like 2 cells; the top 4 terms in the cell compartments (CC) terms are displayed. **E** Percent makeup distributions of each identified cell type across all sampled epithelial cells per subject within each condition. Each dot represents a single subject. Adj.*p*-value calculated in REACTOME with Benjamini-Hochberg. **F** Violin plots of expression of aberrant basaloid cell markers and Lysyl oxidase (LOX) family genes. **G** Differential gene expression of PCLS treated with fibrotic cocktail and verteporfin compared with fibrotic cocktail alone in the aberrant basaloid like 1 cluster (adjusted *p*-value < 0.05, Bonferroni correction in FindMarkers()), negative fold change > −0.5 (blue) ie Downregulated, and positive fold change >0.5 (red) ie Upregulated. **H** Feature-plots of LOX genes showing gene expression across all nuclei in all samples. Source data for Fig. 6 are provided at https://doi.org/10.5281/zenodo.14229565.

incompletely understood which combinations of Yap and/or Taz transcription factor complexes promote regeneration versus which promote further injury or arrested regeneration. VP binds to TEAD and thus might inhibit both YAP-TEAD as well as TAZ-TEAD interactions. However, VP has been primarily reported to act on YAP interactions, based on affinity and protein expression levels in a given cell[83–86]. Thus, loss of Yap-TEAD interactions may promote Taz mediated regeneration. Notably, previous data and our scRNA-Seq as well as snRNA-Seq show that AT2 cells and basaloid cells have a high abundance of YAP expression, with fewer cells expressing TAZ and also at lower levels. This indicates YAP-TEAD inhibition as a prominent mechanism; however, the number of overall TAZ positive fibrotic cells was low, and future studies are needed to further investigate whether this is a direct effect or secondary to YAP-TEAD inhibition. Furthermore, YAP and TAZ have been shown to be regulated by, as well as capable of regulating, diverse developmental pathways which are known to be reactivated in IPF. This includes but is not limited to Sonic hedgehog signaling[56], Wnt[10] and TGF-beta[52,57]. Future work should explore the complex interplay of epithelial YAP and TAZ with these pathways in the context of pulmonary fibrosis.

Our data support YAP-TEAD inhibition by VP as a potential therapeutic option for IPF. In support of this, YAP-TEAD inhibition by VP has been shown to be beneficial in liver and cardiac fibrosis as well as in skin fibroblast of systemic sclerosis[87–89]. Our data indicate that YAP activation in fibrosis might occur early in the disease and contributes to ECM deposition and stiffness, which in turn activates YAP signaling, thus developing a potential vicious feed-forward loop[90,91]. Targeting YAP also opens the potential opportunity to interfere early and stop and slow disease progression, however, further studies are needed to understand the temporal regulation and activity of YAP in disease. It is worth mentioning that other YAP inhibitors should be explored and that Metformin, which has been shown to elicit antifibrotic effects in the lung[92,93] has been reported to target YAP activity[94].

In conclusion, we identify the YAP-TEAD/LOX axis as a epithelial cell-driven pro-fibrotic mechanism, resulting in phenotypic cellular changes as well as ECM alterations, as a pharmacological approach to treat tissue fibrosis.

## Methods
### Study design
The aim of the study was to assess and investigate how fibrotic AT2 cells contribute to pulmonary fibrosis and specifically analyze the role of increased YAP activity in fibrotic AT2 cells. We used paraffin-embedded tissue from experimental lung fibrosis and human IPF and donor tissue from patients to assess YAP expression in situ. Human lung tissue samples were obtained from the CPC-M bioArchive at the Comprehensive Pneumology Center (CPC Munich, Germany) and the Tissue Repository at the University of Pittsburgh. Mice were randomly allocated into experimental groups prior to the onset of any experimentation. For experimental lung fibrosis, mice were subjected to a single dose of bleomycin (Sigma B5507) administered intratracheally

via a microsprayer at 2U/kg body weight. The use of animals was approved under the ethics of the Helmholtz-Zentrum Munich and the state of Bavaria, Germany; project number: 55.2-1-54-2532-88-12. Reporting of animal experiments conforms to ARRIVE 2.0 guidelines. Mice were housed under 12-h light/12-h dark cycles at room temperature (20–24 °C) and humidity (45–65%). Animals were given free access to water and standard chow. 12-week-old female C57BL/6 mice were used for all animal experiments as indicated. Animals were weighed every day to monitor their health; animals which lost more than 15% body weight from the study onset were euthanized.

We isolated primary mouse AT2 cells from fibrotic (14 days after in vivo bleomycin application) and healthy lungs (14 days after in vivo PBS application) as outlined in detail below and subjected them to in vitro cultures, including siRNA-mediated loss-of-function studies according to published protocols[21,95–97]. Supernatants from AT2 cell cultures and BALF samples were subjected to an in vitro collagen fibril formation and gelation assay to investigate whether soluble factors, such as LOX, contribute to ECM crosslinking and deposition.

We further aimed to investigate the potential therapeutic effects of YAP inhibition in vivo and ex vivo. To this end, we used verteporfin as a known YAP-TEAD inhibitor in different in vivo and ex vivo models. For experimental lung fibrosis in vivo, we subjected mice to the well-known bleomycin model and used a therapeutic treatment regime (treatment from day 7 to day 14 after bleomycin). Fibrosis was assessed using qualitative and quantitative gold-standard readouts using a variety of molecular biological assays[98], as outlined below.

Fresh lung tissue of explanted Donor or IPF lungs was used for human PCLS according to previous published protocols[40,41,99]. Donor lung samples were obtained from the Center for Organ Recovery and Education (CORE) at the University of Pittsburgh. Donor lung samples originated from lungs deemed unsuitable for organ transplantation. For the fibrosis induction in hPCLS, PCLS were treated for 5 days with a control cocktail (CC) including all vehicles or a pro-fibrotic cocktail (FC) consisting of TGF-β (5 ng/ml, Bio-Techne), PDGF-AB (10 ng/ml, Thermo Fisher), TNF-α (10 ng/ml, Bio-Techne), and LPA (5 μM, Cayman chemical)[40,42]. For verteporfin treatments, hPCLS were treated with FC or CC to allow for induction of fibrosis and Verteporfin treatment started at day 3 until day 5. At the end of the experiment, PCLS were snap-frozen individually in liquid nitrogen for RNA or protein extractions as well as single nuclei analysis, as described below. The study was approved by the local ethics committee of the Ludwig-Maximilians University of Munich, Germany (Ethic vote #19-630) and University of Pittsburgh (IRB PRO14010265). Written informed consent was obtained for all study participants.

### Immunohistochemistry (IHC) and immunofluorescence (IF)
Human lung tissue was placed in 4% (w/v) paraformaldehyde after explantation (fixation was done for 12–24 h) and processed for paraffin embedding. Sections (3 μm) were cut and mounted on positively charged glass slides (Super Frost Plus, Langenbrinck (Emmendingen, Germany)). Paraffin-embedded tissue sections of normal donor and

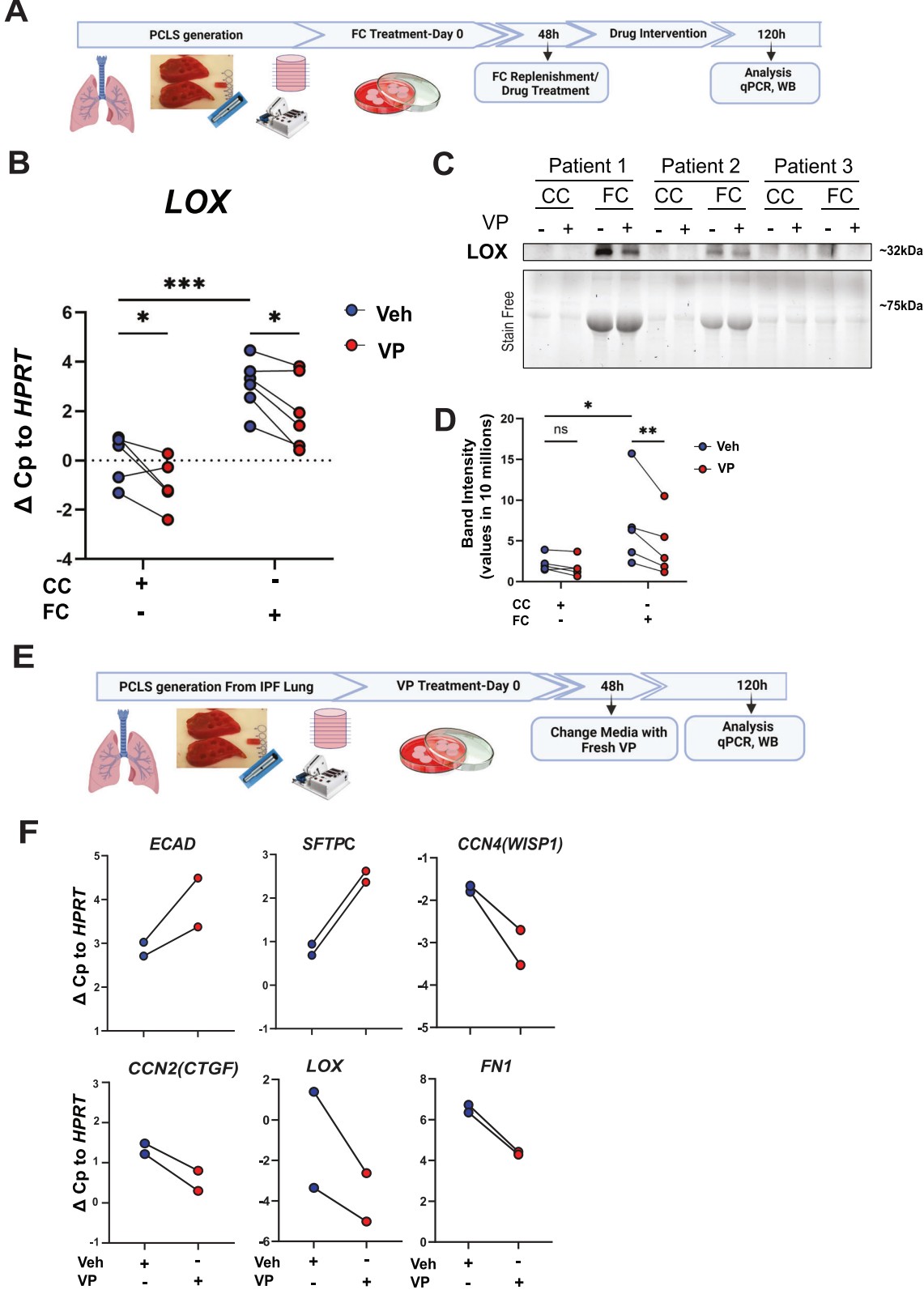

IPF lungs were deparaffinized in xylene and rehydrated in graded alcohol. Antigens were retrieved by cooking the sections for 5 min in 10 mmol/L citrate buffer (pH 6.0) using microwave irradiation (800 W). Thereafter, sections cooled down for 20 min at RT, followed by repeated cooking (800 W, 5 min) and cooling (20 min at RT). This procedure was performed three times. Importantly, the citrate buffer was freshly prepared by mixing 18 mL 100 mmol/L citric acid monohydrate and 82 mL 100 mmol/L sodium citrate tribasic dihydrate with 900 mL distilled water. For immunostaining, the streptavidin-biotin-alkaline phosphatase (AP) method with use of the ZytoChem-Plus AP Kit (Fast Red) [Zytomed Systems, Berlin, Germany], according to the manufacturer´s protocol, was employed.

In general, sections were incubated for 2 h at RT with primary antibodies, which were diluted in PBS containing 2% (w/v) BSA. Control

**Fig. 7 | YAP-TEAD inhibition by verteporfin reduces LOX and fibrotic markers in human fibrotic lung tissue ex vivo. A** Schematics for precision-cut lung slice (PCLS) generation from donor lung and treatment with the fibrosis cocktail (FC) and verteporfin (VP) ex vivo for 5 days. **B** *LOX* gene expression of in PCLS treated with the fibrosis cocktail and VP, $n = 5$ patients for CC and $n = 6$ for FC conditions; Two-way ANOVA with uncorrected Fisher's LSD. CC (Veh (DMSO) vs. VP, *$p = 0.0199$), FC (Veh vs. VP, *$p = 0.0176$), Veh (CC vs. FC, ***$p = 0.0005$). **C**, **D** Immunoblotting and quantification of LOX in supernatants obtained from PCLS in study described in Fig. 7A; (*$p < 0.05$, paired t-test and ns unpaired t-test). Stain-Free technology is shown for qualitative evaluation only as an estimation of total protein in supernatants; it is not used for quantification in (**D**) as it labels tryptophan amino acids which are not present in collagen and thus underestimates

total protein amounts in supernatants (see Supplementary Fig. S17 legend for further details). Quantification in (**D**) was performed on the LOX detected in the separate blots, run in parallel, shown in Panel (**C**) and Supplementary Fig. S17. PCLS derived from individual patients were run on the same blots to permit relative comparisons for the patient's own control conditions. *$p < 0.05$ as assessed by Two-way ANOVA with uncorrected Fisher's LSD. CC (DMSO vs. VP, $p = 0.4421$), FC (DMSO vs. VP, $p = 0.0022$), DMSO (CC vs. FC, $p = 0.0399$); $n = 5$ patients. **E** Schematics for experiments using PCLS derived from IPF tissue explants and treated with VP. **F** Gene expression of *CDH1 (ECAD)*, *SFTPC*, *CCN4 (WISP1)*, *CCN2 (CTGF)*, *LOX*, and *FN1* in IPF PCLS treated with VP, $n = 2$. Panels (**A** and **E**) Created in BioRender. Wagner, D. (2025) https://BioRender.com/nmwzx11. Source data for Panels (**B**–**D**, **F**) are provided as a Source Data file.

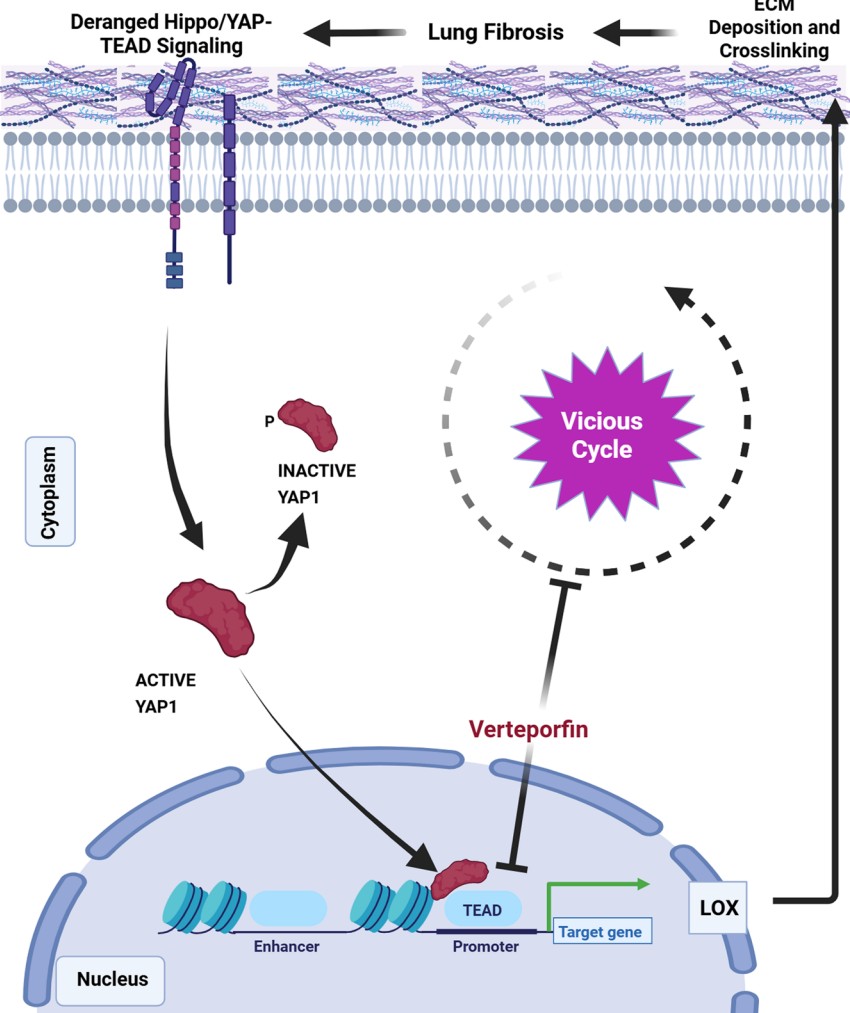

**Fig. 8 | Summary of proposed vicious Hippo-YAP/LOX mediated extracellular matrix (ECM) remodeling cycle driven by lung epithelial cells in IPF.** Derangement of the Hippo signaling pathway components lead to the activation of YAP-TEAD activity in the nucleus leading to increased gene expression of matrix molecules and modifying enzymes (such as LOX) leading to increased secretion of

LOX from the fibrotic alveolar epithelial cells, which contributes to increased crosslinking of secreted matrix in the ECM. This increase in matrix crosslinking causes further changes in the mechanical stiffness that leads to further increase in YAP activity. Created in BioRender. Wagner, D. (2025) https://BioRender.com/nmwzx11.

sections were treated with PBS-2%BSA alone to determine the specificity of the staining. Detection was performed with a polyvalent secondary biotinylated antibody (rabbit, mouse, rat, guinea pig, provided by the ZytoChem- Plus AP Kit, 20 min incubation) followed by incubation with AP-conjugated streptavidin (20 min). Sections were then developed with Fast Red substrate solution, and the reaction was terminated by washing in distilled water. The stained sections were counterstained with hemalaun (Mayers hemalaun solution, WALDECK Division CHROMA GmbH & CO KG, Münster, Germany) and mounted

in Glycergel (DakoCytomation). Lung tissue sections were scanned with a scanning device (Nano-Zoomer, Hamamatsu), and examined histopathologically using the ´NDP.view2 software´ at 100×, 200×, 400× and 800× original magnification.

Immunofluorescence of human lung histological sections was performed similarly as above except for the usage of secondary antibodies conjugated to fluorophores. Whole slides were scanned with an Olympus IX83 P2ZF using a 20× UPLXAPO objective with an NA of 0.8 and images were exported using the Olympus OlyVia 3.1 Software.

Immunofluorescence of murine lung sections was performed as above and imaged with a Nikon A1+ Confocal. All primary and secondary antibodies and their concentrations are included in Supplementary Table 6.

## Histology of murine lungs

Following ligation and surgical resection of the right lung lobes, left lung lobes were gravity-fixed via tracheal inflation with fixation solution (6% Paraformaldehyde (PFA)) for 10 min at room temperature followed by submersion in fixative over night at 4 °C. Individual lobes were then cut transversally into four sections and subjected to standard tissue processing and paraffin embedding. 5um sections were prepared using a microtome (Zeiss) and mounted on glass microscope slides. Hematoxylin and Eosin (HE) and Masson's trichrome (MTC) staining were performed using standard protocols and then scanned using a MIRAX Scan (Zeiss) with Control Software v0.5.

## Modified Ashcroft scoring

The fibrotic state of murine lungs was evaluated using the modified Ashcroft scoring[100]. Briefly, low and higher power magnification images were obtained from MTC stained and scanned murine lungs (MIRAX Viewer Software v.1.12.25.3). All samples were de-identified and randomized. Qualitative evaluation of the images was performed by three independent, blinded experts, scoring each image from a scale of 0–8 where 0 is normal lung to 8 being most fibrotic. Scores for each mouse were averaged across the three independent reviewers and samples were re-identified for further analysis.

## Collagen quantification

Individual right lobes from each animal were used for analysis of hydroxyproline amounts by reverse-phase high performance liquid chromatography (HPLC) as a measure of collagen content. Amounts of hydroxyproline per lobe were correlated to total collagen amount using standard curves processed identically[101].

## Primary alveolar epithelial type 2 (AT2) cell isolation

12-week-old male C57BL/6 mice were administered either PBS or Bleomycin as described in the previous section and were sacrificed for cell isolation at day 14. AT2 cells were isolated as previously described using negative selection, resulting in 92–95% surfactant protein C-positive cells[102]. Briefly, mice were anesthetized using ketamine-xylazine mixture and trachea was intubated with a modified 19 G needle (blunt end and grated edges). Bronchoalveolar lavage fluid (BALF) was collected twice with 500 ul PBS supplemented with protease inhibitors. Lungs were perfused with warm 0.9% NaCl through the heart and filled with 1.5 ml Dispase (50 Caseinolytic units/ml, BD Bioscience) and immediately capped with 300 ul 2% agarose and incubated for 45 min. Lungs from 4 mice were pooled for each isolation. Lung lobes were separated and minced using forceps and cells were passed through 100 um, 20 um and 10 um nylon filters. Cells were then plated in petri dishes and incubated at 37 °C for 30 min for negative selection of fibroblasts and macrophages. Unattached cells were collected and negative selection of CD45+/CD31+ cells was performed using MACs magnetic microbeads and separators. The remaining cells were plated in cell culture plates or on coverslips coated with Poly-L-lysine. Incubation duration and biological replicates are indicated in each experiment. DAEC complete medium consisted of high-glucose DMEM (D0822, Sigma) with 1% Pencillin/streptomycin, 1X glutaMax.

## AT2 cell culture, transfection and treatments

Freshly isolated murine AT2 cells were seeded on tissue culture plastic and allowed to adhere and form epithelial monolayers for two days[103]. Cells were then transfected overnight with Lipofectamine RNAiMax (Invitrogen) according to the manufacturer's instructions using 60 nM

siRNA for Yap and Taz, (Yap, Taz from Santa Cruz: sc-38638, sc-38569), 30 nM for siLox (Integrated DNA Technologies, #488693007) and scrambled siRNA control (Santa Cruz or Origene; SR30004). Media was then changed to AT2 cell complete medium and harvested at 48 h after transfection. For verteporfin treatments, AT2 cells were freshly isolated and allowed to attach prior to treatment with 7 uM of verteporfin in DMSO for 24 or 48 h. Supernatants were collected and stored at −80 °C until use.

## AT2 cell Immunofluorescence staining

AT2 cells were seeded on poly-l-lysine treated coverslips. Cells were stopped at day 2 or after 48 h of respective treatment and fixed with ice-cold acetone-methanol (1:1) for 10 min and washed 3 times with 0.1% BSA in PBS. Next, cells were permeabilized with 0.1% Triton X-100 solution in PBS for 20 min, blocked with 5% BSA in PBS for 30 min at room temperature and incubated with primary antibodies (see Supplementary Table 4) followed by secondary antibodies, 1 h each. DAPI (Roche, Basel, Switzerland) staining for 10 min was used to visualize cell nuclei. Next, coverslips were fixed with 4% PFA for 10 min, mounted with fluorescent mounting medium (Dako, Glostrup, Denmark) and visualized with an Axio Imager M2 Microscope (Zeiss) or confocal microscope (LSM 710; Zeiss).

## RNA isolation and reverse transcription quantitative polymerase chain reaction (RT-qPCR)

Standard nucleic acid extraction protocols were used in obtaining RNA from monolayered cells as previously described using peqGold Total RNA kit (Peqlab, Erlangen, Germany)[97,103,104].

Lung lobes extracted immediately following termination of all in vivo experiments were snapfrozen in liquid nitrogen and stored at −80 °C until further processing. RNA was isolated by mechanically homogenizing tissue using the Mikro Dismembrator S (Sartorius Group, Germany). RNA isolation from pulverized mouse lung tissue was performed using TRIZol followed by purification using peqGOLD columns.

RNA isolation from human PCLS was done using RNeasy Fibrous Tissue Mini Kit (Qiagen, Germany) and Quick RNA Microprep kit (R1051). RNeasy Fibrous Tissue Mini Kit (Qiagen, Germany) was used with minor modifications. Pulverized human PCLS were incubated in RLT lysis buffer containing 4 mM Dithiothreitol (DTT) for 10 min on ice. Proteinase K was added and incubated at room temperature for 15 min. Lysates were passed through DNA removal columns and loaded on RNeasy mini columns and processed further per the manufacturer's instructions. For Quick RNA Microprep kit (R1051), after homogenization of the hPCLS in lysis buffer we followed the manufacturer's instructions.

RNA concentrations and purity were assessed using NanoDrop 1000 (Thermo Fischer Scientific, Germany). cDNA synthesis was prepared using SuperScriptTM II (Invitrogen, Carlsbad, CA, USA). qPCR reactions were performed with the primers (Eurofin Genomics and IDT) listed in Supplementary Tables 2, 3 and SYBR green Master Mix (Roche, Germany). Primers were used at a final concentration of 500 nM and qPCR reactions were performed on an LC light cycler 480 (Roche). $\Delta Cp = Cp(HPRT) - Cp(gene)$.

## Western blotting

Protein lysis in most experiments was done using a modified RIPA buffer containing 50 mM Tris-HCl, 150 mM NaCl, 1 mM egtazic acid (EGTA), 1 mM ethylenediaminetetraacetic acid (EDTA), 1% NP-40, 2.5 mM Tetrasodium pyrophosphate, and 1% sodium deoxycholate (SDC). Mouse lung lobes, mouse PCLS, and human PCLS were snap frozen in liquid nitrogen and pulverized using mikro dismembrator S (Sartorius Group, Germany). Samples were then lysed in RIPA buffer for 30 min on ice and centrifugated at 15,000 × g for 20 min. Protein lysis of monolayered cell cultures was done by directly applying RIPA

lysis buffer and shaking on ice for 15 min. Supernatant samples and bronchioalveolar lavage (BAL) were concentrated by freeze-drying and reconstitution in lysis buffer at a 10X concentration. Samples were denatured in Laemelli buffer at 95 °C with or without beta-mercaptoethanol. Gel electrophoresis was done using Tris based gels at ranging concentrations (6–15%) with the addition of 2,2,2-tri-chloroethanol (TCE) to enable Stain-Free visualization all protein lanes of the gel (Bio-Rad). Prior to transfer, gels were activated using UV for 45 s to detect full protein lanes using fluorescence detection. PVDF and nitrocellulose membranes were used for blotting and blocking was done using 5% non-fat dried milk in Tris buffered saline – Tween 20 (TBST). Primary antibodies were incubated at 4 °C overnight and secondary antibodies were incubated at room temperature for 1 h unless stated otherwise. Protein bands were visualized using Biorad's ChemiDOC XPS (Biorad) and quantified using ImageLab v6.0 (Biorad)

### In vitro cell-free collagen gelation assay
Primary cell culture supernatants were used to study their ability to alter collagen assembly. All samples and their controls were collected in equal volumes, freeze-dried and reconstituted in the same volume of carbonate buffer as previously described[105]. Rat-tail type I collagen (Corning) was diluted in carbonate buffer at 1 mg/ml, pH 7.8 and kept on ice. The biological fluid was then added and immediately mixed well manually. Samples were then incubated at 37 °C and monitored with optical density measurements for the first two hours at 350 nm (Cytation5, Gen5 Software). Samples were then incubated for a further 48 h to allow gelation to fully complete and the resulting gel was analyzed by immunoblotting or fixed in glutaraldehyde/formaldehyde for analysis with a Jeol JSM-7800F scanning electron microscope (SEM) as previously described[106–108]. Images were analyzed using the TWOMBLI v1 tool for lacunarity and fractal dimensions[109].

### Microarray analysis
RNA quality control was performed using High Sensitivity RNA ScreenTape for an Agilent TapeStation. All samples had RIN > 9.0. Transcriptomic analysis was performed using Clariom™ D mouse microarrays. The expression data were processed using the R package oligo(1) in Bioconductor (http://www.bioconductor.org/), including background correction, quantile normalization, log2 transformed and final probe summarization. Using Bioconductor package limma[110], the gene expression of each comparison were compared using a linear model fit followed by Empirical Bayes statistical tests[111]. The results are reported with Log2FC and adjusted *P*-value using Benjamini–Hochberg multiple-test adjustment method implemented in limma package. All bioinformatic analyses were performed using R. Principle component analysis (PCA) was performed on the essential R function *princomp*. Graphing packages used included *ggplot2*, *plotly*, and *pheatmap*. Matrix design was performed in a paired manner between the different siRNA treatments on AT2 cells. Contrast matrix included comparisons between all conditions. Gene lists were produced using a cut off *p*-value of 0.05.

### Fourier Transform Infrared Spectroscopy (FTIR) data collection and processing
5 um lung tissue slices from paraffin-embedded samples were analyzed in transmission mode using a Diamond-ATR accessory for a Cary630 FTIR with KBr optics (Agilent). Spectra were collected from 4000 to 650 cm⁻¹ at a resolution of 2 cm⁻¹ with 8 measurements taken as background and 8 measurements per sample and averaged. Absorbance was calculated using the formula:

Absorbance = 2 − log(%*T*) where *T* is transmittance.

Paraffin spectra were collected and subtracted from all samples. Spectra were smoothened using a Savitzky–Golay filter with order 3 and length 51 prior to background subtraction in R. Quasar version 1.10.2, an open-source toolbox extending the capabilities of Orange

3.37.0 and Orange Spectroscopy v.0.7.2, was used for second derivative spectroscopy to confirm peak identification according to the literature (Supplementary Table 3) prior to extraction of background corrected samples for statistical comparison.

### Sample preparation and nuclei extraction from human PCLS
Four PCLS slices per control donor (*n* = 2) at day 5 after DMSO, FC, FC+ Verteporfin stimulation were washed in cold 1X PBS and snap frozen. Nuclei were extracted using the Nuclei Isolation kit (CG000505, 10X Genomics,). Briefly and based on the manufacturer's protocol and reagents, the tissue was dissociated on ice, centrifugated and washed. The pellet was resuspended, and cellular debris was removed. Following another centrifugation step, nuclei were resuspended and counted.

### Single-cell barcoding, library preparation, and sequencing
Around 20'000 nuclei were loaded on a Chip G with Chromium Single Cell 3' v3.1 gel beads and reagents (3' GEX v3.1, 10x Genomics). Final libraries were analyzed on an Agilent Bioanalyzer High Sensitivity DNA chip for qualitative control purposes. cDNA libraries were sequenced on a HiSeq 4000 Illumina platform aiming for 150 million reads per library and a sequencing configuration of 26 base pair (bp) on read1 and 98 bp on read2.

### Fastq generation and read trimming
Basecalls were converted to reads with the software Cell Ranger's (v4.0.0) implementation mkfastq. Multiple fastq files from the same library and strand were catenated to single files. Read2 files were subject to two passes of contaminant trimming with cutadapt (i) for the template switch oligo sequence (AAGCAGTGGTATCAACGCA-GAGTACATGGG) anchored on the 5' end and (ii) for poly(A) sequences on the 3' end. Following trimming, read pairs were removed if the read2 was trimmed below 30 bp.

Paired reads were filtered if either the cell barcode or unique molecular identifier (UMI) sequence had more than 1 bp with a phred of <20. Reads were aligned with STAR (v2.7.9a) to the human genome reference GRCh38 release 99 from ensemble. Collapsed UMIs with reads that span both exonic and intronic sequences were retained as both separate and combined gene expression assays.

### Filtering cell barcodes and quality control
After preprocessing, analysis of the ex vivo human PCLS snRNA-seq data was conducted using the Seurat package (version 4.1.3). Nucleii with less than 750 transcripts profiled and with a percentage of reads that map to the mitochondrial genome >3% were then removed. The average number of nuclei returned per library is 10,281 (±6517). A detailed summary of the average UMI and gene per cell type per library distribution per treatment condition is presented in Supplementary Fig. S10.

### Integration and analysis
To minimize the possible effect of potential batch correction methods, we first processed and annotated each library separately, before integrating them together and annotating them jointly. To integrate the multiple snRNA-seq datasets, we employed Robust Principal Component Analysis (RPCA). RPCA is a powerful technique for decomposing a data matrix into low-rank and sparse components. Briefly, the low-rank component represents shared biological signals across datasets, while the sparse component captures dataset-specific variations and technical noise. Based on the cellular diversity, we chose to use PCLS treated with DMSO as the reference for the integration. Following the RPCA decomposition, we utilized the low-rank component as the integrated representation of the snRNA-seq datasets. This component captured shared biological signals across conditions while mitigating dataset-specific variations. However, subsequent analyses, such as

clustering and differential expression analysis, were performed on the non-integrated but normalized gene expression values. To validate the effectiveness of the integrated representation, we performed various analyses, including cell-type clustering and identification of marker genes. We also compared the results of these analyses to those obtained from individual datasets to evaluate the improvement gained through the integration process. Marker genes were computed using a Wilcoxon rank-sum test, and genes were considered marker genes if the FDR-corrected p-value was below 0.05 and the log2 fold change was above 0.5.

## Statistical analysis

Data are presented as mean ± SEM or mean ± SD as outlined in the text and figure legends, from n separate experiments. Individual murine lungs represent $n = 1$, AT2 isolates (from 4 to 6 mouse lungs each) were considered $n = 1$, For PCLS, we pool 4–5 slices/punches per biological sample/condition. Data acquired by subjective measures (e.g., scoring histology) was performed blinded. We used T-test, ANOVA, or non-parametric tests, as appropriate and based on the underlying distribution of the data and the number of comparisons performed as indicated in the corresponding figure legend in Graph-Pad Prism v10.4.2 Correlation was evaluated using Pearson's test. Differences were considered to be statistically significant when $p < 0.05$. Mean and standard deviation are shown in all graphs unless otherwise indicated.

## Lung function correlation

Lung function and gene expression correlations were performed by plotting normalized array expression values against % DLCO values extracted from the Lung Genome Research Consortium for normal and IPF patient samples run on Agilent-028004 SurePrint G3 Human GE 8x60K Microarray (GSE47460). Patients without % DLCO reported were excluded.

## Reporting summary

Further information on research design is available in the Nature Portfolio Reporting Summary linked to this article.

## Data availability

The raw data from microarrays performed on normal and fibrotic AT2 cells treated with or without siRNA for Yap/Taz have been deposited in the ArrayExpress database under accession number E-MTAB-14643. All murine histological slides used for Modified Ashcroft scoring, human immunohistochemistry of parallel sections of YAP/TAZ with co-stains, and SEM images of collagen networks for quantification have been deposited in the BioImage Archive S-BIAD1520. 10X Genomics data and metadata generated for this study have been deposited in the Zenodo database (https://doi.org/10.5281/zenodo.14229565)[112]. All other data generated in this study are provided in the Supplementary Information/Source Data file. Source data are provided with this paper.

## Code availability

Code used to generate the various figures in this manuscript can be found on the github repository: https://github.com/Lung-bioengineering-regeneration-lab/Hippo_LOX[113].

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

## Acknowledgements

We gratefully acknowledge the provision of human biomaterial and clinical data from the CPC-M bioArchive and its partners at the Asklepios Biobank Gauting and the Klinikum der Universität München. We thank

Marlene Stein and Anastasia van den Berg for their excellent assistance in this project and all previous and current #PinkLab members for discussions and contributions. This work was supported in part by funds and facilities provided by the Technology Enhancing Cognition and Health Geriatric Research Education and Clinical Center (TECH-GRECC) at the VA Pittsburgh Healthcare System, Pittsburgh, Pennsylvania. The findings and conclusions in this document are those of the authors, who are responsible for its content, and do not necessarily represent the views of the VA or of the United States Government. We acknowledge support from the Lund Bioimaging Center. M.K. acknowledges support by the German Center for Lung Research (DZL) and the Helmholtz Association (W2 Professorship Program). M.K. and N.K. acknowledge funding by the Three Lakes Foundation (Three Lakes Consortium for Pulmonary Fibrosis). M.K. and O.E. acknowledge funding from the National Institute of Health Common Funds (U54 AG075931). O.E. acknowledges funding by the National Institute of Health (NIH R01 HL146519) and the Chan Zuckerberg foundation (Human Lung Cell Atlas CZI25A8523). D.E.W. acknowledges financial support by an American Thoracic Society Unrestricted Grant, the Helmholtz Zentrum München Postdoctoral Fellowship and a Whitaker Foundation International Scholar Award. This project was partially funded by the Swedish Research Council Starting Grant (Dnr 2018–02352) and has received funding from the European Research Council (ERC) under the European Union's Horizon 2020 research and innovation program (3DBIOLUNG, Grant Agreement 805361; to D.E.W.). This research was undertaken, in part, thanks to funding from the Canada Excellence Research Chairs Program (to D.E.W.; CERC-2022-00013). M.L. acknowledges support from the Deutsche Forschungsgemeinschaft (grant agreement number 512453064, BfR 60-0102-01.P588). A.J. acknowledges funding from Breath endowment fund (Fds 2019-Ostinelli).

## Author contributions

D.E.W. and M.K. conceived and designed the study. D.E.W., C.S.W., M.L., A.H., A.G., I.R., N.K., R.Ch., Sd.L., O.E., and M.K. supervised the experimental work. D.E.W., H.N.A., N.M., M.L. and M.K. wrote the paper. D.E.W., H.N.A., N.M., M.Kor., K.W., U.O., A.E.B., M.M., K.M., R.C., D.B., J.St., W.S.W., S.K., C.O., and H.A.B. performed in vitro and in vivo experiments. H.N.A., N.M., J.C.G., A.J., R.P., and J.Se. designed and performed ex vivo PCLS work and snRNA-seq data acquisition. H.N.A., A.J., D.E.W. and Q.H. performed bioinformatic analysis of scRNA and snRNA-seq data. N.G. acquired FTIR data and N.G., H.N.A. and D.E.W. performed data analysis. J.D., J.T., and H.N.A. performed spatial transcriptomic analysis. D.E.W., I.R., N.K., O.E., M.K. provided funding and resources. All authors read and corrected the final manuscript.

## Competing interests

The authors declare no competing interests.

## Additional information

[1]Lund Stem Cell Center, Faculty of Medicine, Lund University, Lund, Sweden. [2]Lung Bioengineering and Regeneration, Department of Experimental Medical Sciences, Faculty of Medicine Lund University, Lund, Sweden. [3]Wallenberg Center for Molecular Medicine, Faculty of Medicine, Lund University, Lund, Sweden. [4]NanoLund, Lund University, Lund, Sweden. [5]Lung Repair and Regeneration Research Unit, Helmholtz Zentrum München, Member of the German Center for Lung Research (DZL), Munich, Germany. [6]Meakins-Christie Laboratories, Research-Institute of the McGill University Hospital, Montreal, QC, Canada. [7]Department of Medicine, McGill University, Montreal, QC, Canada. [8]Department of Biomedical Engineering, McGill University, Montreal, QC, Canada. [9]Division of Pulmonary, Allergy, Critical Care, and Sleep Medicine, Department of Medicine University of Pittsburgh, Pittsburgh, PA, USA. [10]Geriatric Research Education and Clinical Center (GRECC) at the VA Pittsburgh Healthcare System, Pittsburgh, PA, USA. [11]Pulmonary, Critical Care and Sleep Medicine, Yale School of Medicine, New Haven, CT, USA. [12]Department of Pulmonary Medicine, Interstitial Lung Disease Center, University Hospital of Caen UNICAEN, Caen Normandie, CNRS, Normandie University, ISTCT, UMR6030, GIP Cyceron, Caen, France. [13]Institute for Lung Health and Immunity, Helmholtz Zentrum München and University Hospital of the Ludwig Maximilians Universität, Member of the German Center for Lung Research (DZL), Munich, Germany. [14]Department of Pediatrics and Division of Pulmonary Sciences and Sleep Medicine, University of Colorado, Anschutz Medical Campus, School of Medicine, Aurora, CO, USA. [15]Dept of Internal Medicine, Justus-Liebig-Universität Giessen, Giessen, Germany. [16]Universities of Giessen and Marburg Lung Center (UGMLC), Member of the German Center for Lung Research (DZL), Giessen, Germany. [17]Department of Internal Medicine. Yale University, New Haven, CT, USA. [18]Section of Pulmonary, Critical Care, and Sleep Medicine, Department of Medicine, Baylor College of Medicine, Houston, TX, USA. [19]Centre for Inflammation and Tissue Repair, UCL Respiratory, University College London, London, UK. [20]Department of Medicine, Division of Pulmonary and Critical Medicine, Mayo Clinic, Rochester, NY, USA. [21]Institute for Lung Research, Philipps-University Marburg, Marburg, Germany. [22]Institute for Lung Health, German Center for Lung Research (DZL), Giessen, Germany. ✉e-mail: darcy.wagner@mcgill.ca; koenigshoffm@upmc.edu

