## [Transparent Peer Review file · Nature Communications]

Inhibition of epithelial cell YAP-TEAD/LOX signaling attenuates pulmonary fibrosis in preclinical models

Corresponding Author: Professor Melanie Königshoff

Version 0:

Reviewer comments:

Reviewer #1

(Remarks to the Author)

This manuscript seeks to determine the role of Yap activity in AT2 cells in the activation of the fibroblasts population in the IPF lung, and identifies a Yap/TEAD/Lox signaling axis that is associated with enhanced collagen deposition. Inhibition of this axis with either Verteporfin or siYT inhibits Lox and downstream enhanced collagen assembly. In general these findings are well written. Some of the findings are in need of further clarification to support the authors claims.

Figure 5 is very compelling

In several instances, i.e. Figure 1C, 2A, 4C the claims of Yap/Taz or Lox in AT2 cells would be greatly supported by co-staining with AT2 markers such as Sp-C, Abca3, or Lamp. The Yap/Taz antibody stain is unclear whether it is in the epithelium, or mesenchyme- or if it is in AT2 cells or AT1 cells (which some of the authors and others have shown to be high in Taz).

This co-staining would help define whether Yap is nuclear in the AT2 cells. For figure 4C E-Cad would be sufficient to show Lox in At2/epithelium.

In figure 2C, there appear to be a different number of N's in the different targets in the Bleo+VP group ranging from 13-16 samples (Dots).

Figure 3C, did siYT have any effect of cells treated in PBS?

The role of the epithelium and collagen deposition, it is worth noting that increasing evidence implicates the AT1 cell in generating signals associated with basement membrane. Might this play a role in the increased signals when the At2 cells are plated on a dish as these cells tend to differentiate towards an AT1 like state?

As previous work has shown that Lox is regulated by Tgfb, while this situation it appears not to be, should some discussion be given to the complex interplay between Yap/Taz/Tgfb?

Is there evidence that Tead is required for this Yap to Lox interaction? The verteporfin which classically blocks Yap/Tead interactions has also been shown to enhance Yap degradation or other potential mechanisms. It is also interesting in the human lung slices that Vp appears to enhance the number of aberrant basaloid cells (though clearly not significant it is interesting it wasnt reduced) while still reducing the pro-fibrotic signal.

Figure 7E, it appears N=2 for these experiments- additional N would allow for statistical interpretation.

Discussion starting at 387, about the specificity of Verteporfin- would it not be preferred to inhibit Yap in both the epithelium that this paper focuses on, but also in other cell types such as the mesenchyme, that other groups have shown that Yap aberrant regulates? Is there a cell type that lung specific inhibition of Yap in this context not be adaptive?

Minor comments:

Figure 2 C Acta graph is mis-spelled.

Line 124 should Let->led?

line 266 eproduce-> produce

Were the Verteporfin experiments protected from light? As previous reports have shown it to be effected by exposure.

line 322, this paper also showed Yap activation in AT2 cells.

Reviewer #2

(Remarks to the Author)

The manuscript titled “Inhibition of epithelial cell YAP-TEAD/LOX signaling attenuates pulmonary fibrosis” by Wagner et al presents findings on the role of YAP-TEAD signaling in pulmonary fibrosis. The study demonstrates that YAP activity leads to increased expression of LOX in fibrotic AT2 cells. Pharmacological YAP inhibition or YAP siRNA reduced LOX expression, and LOX inhibition reduced lung fibrosis in mice and in human fibrotic tissue. In conclusion, the authors highlight the potential therapy for targeting YAP-TEAD/LOX pathway for treating IPF patients.

However, while this study provides intriguing insights into the regulation of LOX expression by YAP, the impact of YAP loss in AT2 cells on reducing lung fibrosis remains controversial. Previous studies have reported conflicting results, with some showing that loss of YAP leads to reduced differentiation of AT2 cells into AT1 cells and increased lung fibrosis, while others have observed the opposite effect. The underlying reasons for this discrepancy are still unknown, making it unclear whether YAP loss-of-function is ultimately beneficial or detrimental in the context of lung fibrosis.

In addition, this study focuses on the role of YAP-TEAD in promoting ECM stabilization via LOX. However, the role of YAP-TEAD activation in regulating ECM degradation, including by modulating matrix metalloproteinase (MMP), is well-documented, especially in the cancer field. YAP-TEAD signaling has been implicated in the regulation of MMP expression in various contexts, including cancer progression and tissue remodeling. YAP-TEAD can directly bind to the promoters of MMP genes and regulate their transcription. Thus, it is unclear how YAP-TEAD signaling balances the signals between pro- and anti-ECM stability and maturation.

Lastly, AT2 cells are known as alveolar stem or progenitor cells. Their activity, including proliferation and differentiation, is critical for lung repair. It is unclear what effects LOX inhibition has on AT2 cell activity as well as senescence. While this study may provide interesting findings on the role of YAP in AT2 cells in influencing ECM, the lack of scientific rationale and comprehensive characterization of YAP-TEAD activity in ECM remodeling in the setting of lung fibrosis reduces the enthusiasm toward to this study.

Reviewer #3

(Remarks to the Author)

In this study Wagner et al. delve into the analysis of the in vivo role of YAP signaling in lung fibrosis, focusing on the role of AT2 cells regulating extracellular matrix (ECM) remodeling. Further, the authors addressed the potential usefulness of targeting the axis YAP-TEAD/LOX as a therapeutic approach for idiopathic pulmonary fibrosis (IPF). They find that YAP signaling is early activated in fibrotic AT2 in lung fibrosis, and that pharmacological YAP inhibition by verteporfin reverses fibrotic AT2 cell reprogramming and LOX expression in experimental lung fibrosis, in vivo, and in human fibrotic tissue ex vivo. They conclude that YAP-TEAD/LOX inhibition in AT2 cells is a promising new therapy for IPF patients. The study is interesting, but it should be taken into account that the YAP dependent regulation of LOX as well as the benefit of verteporfin on fibrotic diseases have been previously reported.

Major Comments

1. The regulatory feedback loop between YAP signaling, LOX isoenzymes and ECM remodeling has been previously described in other pathological scenarios such as pulmonary hypertension (Bertero T. et al. Cell Rep. 2015;13:1016-32) or liver fibrosis (Cheng F. et al. J Nanobiotechnology. 2023;21(1):195). These or other references on this issue should be included in the manuscript, and some categorical statements such as “... this newly identified YAP-TEAD/LOX axis ...” should be tempered. Authors should clearly state that the novelty of their findings specifically refers to AT2 cells.
2. Introduction (last paragraph), Results and Discussion. Authors state that YAP regulates LOX expression and secretion. Because the secretion of LOX does not appear to have been directly analyzed, authors should describe their findings more properly.
3. Figure 2 C. LOX and other LOXLs should be analyzed. The authors assume that LOX is the main isoenzyme involved in IPF but there are conflicting data in the literature. Which is the contribution of each isoenzyme to the global LOX activity?
4. In any of the studies/samples used, and in particular in human samples, did the authors observe colocalization of LOX with YAP?
5. Figure 2F. Crosslinked collagen should be analyzed using specific techniques (e.g. LC-MS/MS or second-harmonic generation microscopy).
6. Figure 3. LOXLs should be analyzed in panel D. Immunoblot data shown in Figure 3E are not convincing due to the low number of samples. Further, LOX activity should be assessed to confirm the impact of siYAP.
7. Figure 4. Data corresponding to Figure 4B and 4E should be confirmed at the level of LOX activity or collagen crosslinking.
8. Figure 5. Data from experiments using BAPN, should be confirmed in LOX knockdown experiments. As indicated above a more reliable method to assess crosslinked collagen should be used (panel E).
9. In many studies the number of replicates per experiment is low: in many cases only 3 per group, and in immunoblots only

2 or 1 lanes/samples per experimental condition are shown.

Minor points

1. Supplementary Figure S1 only consists of one panel, therefore S1A does not exist.
2. Figure 7A is referred in the text before Figure 6A. Panel 7A should be included in Figure 6 for easy reading.
3. All tables should be referred as "supplementary table" and numbered according to its order of appearance in the text.
4. The sentence in line 292 is speculative since ECM stiffness was not measured.
5. In the immunoblot shown in Figure 7C a loading control should be included.
6. Discussion. The assumption that LOX is not induced by TGFbeta (lines 365-366) is not correct. This issue is known from 90's (please see the review by Laczko and Csiszar Biomolecules 2020).
7. Please check spelling through the manuscript (e.g. limited, interstingly,)

Version 1:

Reviewer comments:

Reviewer #1

(Remarks to the Author)

This manuscript from Wagner et al/ the Konigshoff group is well written and demonstrates that activation of Yap in IPF AT2 cells is associated with activation of Lox and subsequent fibrotic remodeling. Inhibition of this Yap/Lox axis with the Yap/Tead inhibitor Verteporfin reduces fibrotic remodeling in mouse models of fibrosis at 14 days, and reduces fibrotic signaling in human lung sections. The data strongly supports these concepts, and the revisions are well done and greatly enhance the manuscript. My concerns were thoroughly addressed.

Reviewer #2

(Remarks to the Author)

1. The new data that are used to address reviewers comments, including "Confidential Reviewer 5", should be provided in the revised manuscript.

2. Figures 6 and 7 provided interesting data on the effects of VP on lung fibrosis. However, a critical question remains unanswered: which cell type is the primary target of VP's action? Both fibroblasts and epithelial cells can produce LOX. Unfortunately, this study does not distinguish between VP's effects on these cell types, leaving uncertainty about which cell type contributes most significantly to the observed anti-fibrotic effects.

3. Figure 2A showed that doses of 2 U/kg bleomycin caused about 37% mice lethality. However, based on the published research results, 2 U/kg bleomycin treatment typically does not cause mouse lethality.

4. Figure 3H: The co-localization of LOX, HTII-280, and YAP is not clear.

5. Figure 7C: It's unclear if LOX expression was really increased in FC, since the total protein loading for FC appeared to be higher than CC. This makes the quantitative interpretation unreliable.

Reviewer #3

(Remarks to the Author)

No further questions

Version 2:

Reviewer comments:

Reviewer #2

(Remarks to the Author)

Thank you for addressing my concerns. While a brief clarification on collagen's effect on stain-free gel normalization would strengthen the LOX comparison between FC and CC, the findings on VP inhibition remain compelling. I support publication.

REVIEWER COMMENTS

Reviewer #1 (Remarks to the Author):

R1-1. This manuscript seeks to determine the role of Yap activity in AT2 cells in the activation of the fibroblast's population in the IPF lung and identifies a Yap/TEAD/Lox signaling axis that is associated with enhanced collagen deposition. Inhibition of this axis with either Verteporfin or siYT inhibits Lox and downstream enhanced collagen assembly. In general, these findings are well written. Some of the findings are in need of further clarification to support the authors' claims. Figure 5 is very compelling.

C1-1. We thank the reviewer for the positive comments and instructive suggestions. We have now addressed all questions and comments in the revised version of our manuscript, which we believe benefited greatly in response to the reviewer comments. In short, we have performed a significant number of additional experiments to further support and corroborate our findings. These new data include additional immunofluorescence staining of the YAP/LOX axis in AT2 cells in mouse models (new Fig. 1c, 2b, and 4c) and human IPF (new Fig. 3H) as well as additional data which confirms changes in collagen formation and crosslinking with verteporfin administration (Fig. 5h). We further significantly revised the manuscript text, in particular the discussion to include other points raised by the reviewer(s) in relation to the current state of knowledge of YAP/TAZ signaling and the interplay with other signaling pathways in the context of lung injury.

R1-2. In several instances, i.e. Figure 1C, 2A, 4C the claims of Yap/Taz or Lox in AT2 cells would be greatly supported by co-staining with AT2 markers such as Sp-C, Abca3, or Lamp. The Yap/Taz antibody stain is unclear whether it is in the epithelium, or mesenchyme- or if it is in AT2 cells or AT1 cells (which some of the authors and others have shown to be high in Taz). This co-staining would help define whether Yap is nuclear in the AT2 cells. For figure 4C E-Cad would be sufficient to show Lox in At2/epithelium.

C1-2. We thank the reviewer for this suggestion and agree that this additional analysis would be useful. We have therefore performed new co-stainings of Yap/Taz or Lox with DC-Lamp as a marker of AT2 cells in all requested experiments (see new Figures 1c, 2b, and 4c). In addition, we have performed staining of human IPF and Donor lungs with LOX and HTII-280 (a well-accepted marker of AT2 cells) which can now be found in Figure 3h.

The reviewer is correct that there are several papers showing staining of Yap/Taz in mesenchymal cells. However, in our hands, we see Yap strongly expressed in the epithelial compartment in normal and diseased human lung tissue (Figure 1), further corroborated by our new experiments (Figure 3h). Nonetheless, we fully agree that mesenchymal cells also express YAP/TAZ. We have added this to our discussion as well as a recent paper which showed evidence of dysregulated YAP/TAZ in the endothelium in the context of aging associated fibrosis¹. (Lines 422-424) Here, we focus on dissecting the role of YAP/TAZ in the

distal epithelium in the context of fibrosis and thus focused on providing evidence of expression in these cells. In addition, we added spatial transcriptomic data which became publicly available during the revision period². The use of spatial transcriptomic data avoids the potential bias encountered in single cell sequencing experiments where immune cells are known to be overrepresented while AT1 cells are underrepresented. By mining this dataset, we find upregulation of *LOX* in multiple distal epithelial cells in IPF as compared to healthy control tissue samples, providing further evidence that *LOX* is expressed in multiple epithelial cells (new Figure 3G). In addition, we have further analyzed our snRNAseq data which covers the epithelial landscape better than scRNAseq and find that *LOX* is indeed expressed by distal epithelial cells in human PCLS and that this is modulated with verteporfin (Figure 6 and S11).

R1-3. In figure 2C, there appear to be a different number of N's in the different targets in the Bleo+VP group ranging from 13-16 samples (Dots).

C1-3. We apologize for this confusion. We refer the reviewer to the revised panel 2A which we hope more clearly shows that we had differential survival rates among the different groups. Based on our previous work and a power analysis, we determined that 10 mice in the PBS groups and 16 mice were needed in our bleomycin group to ensure sufficient n's survived to determine statistical significance with our chosen endpoints. Due to practical reasons, this was run over two separate cohorts of 26 mice each starting in each experiment (5 in each PBS group and 8 in each bleomycin group). We have added the starting numbers in each group more clearly to the Kaplan-Meier survival curve in the revised version of the figures and have added the survival numbers in the text.

R1-4. Figure 3C, did siYT have any effect of cells treated in PBS?

C1-4. Thank you for this question. We indeed performed these experiments as controls but did not show them in the initial submission. We have now included this data as well as a formal analysis comparing significantly downregulated genes between siYT in both PBS and Bleo treated animals (Figure S4 E-G). Interestingly, we see strong overlap between the genes downregulated (97 genes overlap out of 177 genes downregulated in PBS AT2 cells and 211 genes in Bleo AT2; 83 genes overlap of 175 upregulated in PBS AT2 and 194 BLEO AT2 with Yap/Taz knockdown). This represents around 50-55% overlap between the two conditions. Further, due to question R1-5 and several recent papers showing the role of Yap in transdifferentiation of AT2 to AT1 cells, we looked for genes identified in recent murine single cell analyses that distinguish different alveolar clusters. Interestingly, for siYT-treated PBS AT2 cells, we observed limited evidence of transdifferentiation of AT2 to AT1 (Figure S4 E-G). Under conditions of Yap/Taz knockdown, 5 (of 42) genes associated with AT2 were changed with 4 upregulated (*Dram1*, *Egfl6*, *Sftpb*, and *Ctsc*) and one downregulated (*Lrp2*) indicating retention of the AT2 phenotype. Only 1 gene (of 31) was changed for AT1 cells in

each of the up or downregulated conditions (Akap5). Thus, in our hands, it does not appear that Yap/Taz knockdown in primary AT2s is a master regulator of differentiation of AT2 to AT1 cells. In light of the current literature demonstrating a role of specifically Taz in AT2 to AT1 cell differentiation^{3,4}, this is an interesting finding which warrants further exploration in future work. This may be due to differences between *in vitro* and *in vivo* conditions. We have included these analyses and new figures in Supplement S4.

R1-5. The role of the epithelium and collagen deposition, it is worth noting that increasing evidence implicates the AT1 cell in generating signals associated with basement membrane. Might this play a role in the increased signals when the At2 cells are plated on a dish as these cells tend to differentiate towards an AT1 like state?

C1-5. We thank the reviewer for raising this interesting point. Our findings indeed raise the question whether YAP activity in AT2, but also other cells, such as AT1 cells, impacts other ECM proteins as well, including basement membrane components. Indeed, we found that several basement membrane proteins (Col4 family) were regulated by Yap/Taz in our knockdown experiments. Col4 family members have been recently shown to be secreted by AT1 cells postnatally and to play a major role in AT2 to AT1 differentiation⁷. We hope to address this in future studies and discuss this in the revised version of this manuscript.

We and others have indeed previously shown that plating of AT2 cells onto tissue culture plastic induces AT2 to AT1 differentiation and would like to highlight that all of our experiments have been done with and compared to timepoint matched controls. We observed that *Lox* remains elevated in AT2 cells isolated from mice treated with bleomycin and cultured on tissue culture plastic in comparison to AT2 cells isolated from healthy mice (Figure 3D, first three columns are fibrotic AT2 versus healthy AT2). We have also observed elevated levels of secreted *Lox* protein from AT2 cells derived from animals treated with bleomycin using both western blot and mass spectrometry-based proteomics (Confidential Reviewer Figure 1). However, the mass spectrometry proteomics is planned to be included in a forthcoming manuscript and is shown here for reviewers only.

As discussed in the answer above, new Supplemental Figure S4 does not seem to indicate that in our hands, Yap/Taz play a major role at the transcript level for differentiation of AT2 to AT1. While we acknowledge that a few key phenotypic markers appear in our list of up (e.g. *Sftpb* and *Areg*) or downregulated genes (*Akap5* and *Lrp2*), this is a very small subset of genes in comparison to the genes identified in single cell studies. This may be due to limitations of primary cell monocultures; thus, we have cautiously phrased this in the revised manuscript.

R1-6. As previous work has shown that *Lox* is regulated by *Tgfb*, while this situation appears not to be, should some discussion be given to the complex interplay between Yap/Taz/*Tgfb*?

C1-6. As the reviewer points out, the picture which emerges for the signaling crosstalk between YAP-TAZ and TGF- β is indeed complex but has been shown to be both context and cell-dependent. We have added text to the discussion regarding this aspect and also included a critical reference which identified YAP-TEAD interactions as regulating LOX expression independent of TGF- β modulation.(Lines 464-470 and 533-537)

R1-7. Is there evidence that Tead is required for this Yap to Lox interaction? The verteporfin which classically blocks Yap/Tead interactions has also been shown to enhance Yap degradation or other potential mechanisms.

C1-7. Thank you for raising this important point. While we agree that VP has been shown to potentially target other mechanisms other than YAP/Tead interaction, the large amount of data available is strongly supporting YAP/Tead as the main mechanisms of action. In addition, we have added details on a previous Chip-seq experiment performed by the Guan group which originally identified the TEAD family as being the main transcription factors which interact with Yap⁸. They found that Yap5SA (which promotes TEAD binding) resulted in clear upregulation of LOX and LOXL2. However, when a Yap mutant which cannot bind with TEAD was used (i.e. YapS94A), LOX was no longer upregulated. While performed in breast epithelial cells, this is strong evidence that the YAP-TEAD axis is capable of specifically regulated LOX. As we cannot exclude potential other mechanisms, we included this as a clear limitation in our results and discussion section of the revised manuscript. (Lines 467-470)

R1-8. It is also interesting in the human lung slices that Vp appears to enhance the number of aberrant basaloid cells (though clearly not significant it is interesting it wasn't reduced) while still reducing the pro-fibrotic signal.

C1-8. This is an interesting observation, which we have noted early on as well. Indeed, we do see primarily an effect on the gene expression profile in basaloid cells, which suggests an anti-fibrotic response. Notably, we do have similar data from a larger compound testing cohort in PCLS, in which we tested 10 known and new anti-fibrotic compounds. For all of them, if they target the basaloid cells, we observed effects on gene expression but not on number of total basaloid cells. This indicates a potential limitation of our experimental ex vivo model, especially given that we were only to culture these reliably for up to 5 days. A longer culture time, which would allow for longer drug treatment might result in stronger effects, including effects on basaloid cell numbers.

An alternative explanation would be that only distinct subtypes of basaloid cells are changed, while other (more “healthy” ones) are increasing. In order to work towards addressing this during the revision, we have reanalyzed our snRNAseq data to try and discern further subclusters with which we could perform such an analysis. While we acknowledge the limited number of patients which have been used in our study for snRNAseq, we found that we were able to identify two aberrant basaloid like clusters, which we now term ABL1 and ABL2 as well as two basal cell clusters (basal cells and inflammatory

basal cells). ABL2 appear to arise spontaneously over time in the model while ABL1 are more associated with the FC. Interestingly, and similar to what we showed in the first submission, there is no change in the amount of ABL1 or ABL2 cells (Figure 6E). However, we do see a slight increase in the number of cells in the basal cell cluster in the FC+VP group and virtually no basal cells in the FC group (Figure 6A). This data supports the notion that VP can in fact prevent the acquisition of inflammatory state of basal or ABL cells. We are actively investigating this hypothesis; however, additional sequencing data are needed to draw stronger conclusions.

Nonetheless, we believe that our ex vivo PCLS model is extremely valuable as it allowed us to study living human lung tissue. Despite the short culture time, by using snRNA Sequencing, we were able to show that VP indeed targets basaloid cells, which have been first described in single cell analysis from IPF lungs. This provides clinical and translational relevance for our study.

R1-9. Figure 7E, it appears N=2 for these experiments- additional N would allow for statistical interpretation.

C1-9. We agree with the reviewer and would have loved to be able to repeat this experiment with additional IPF lung tissue. Unfortunately, we did not receive suitable tissue from patients with this rare disease during the revision time and as such were not able to add higher n-numbers to allow for statistical analysis. We have a cohort of human PCLS derived from low to moderate regions of damage in IPF explant tissue which we have treated with VP as well as other compounds (Stegmayr et al. *in preparation*). This is intended for a larger subsequent manuscript and thus we include the data below for reviewer's only which shows that our data here are in line with these additional patients (Confidential Reviewer Figure 2). In the revised manuscript, we make sure to use caution when interpreting these data and mention this accordingly in the revised version of this manuscript. We would like to highlight that we believe that we use FC in 5 patients to induce lung injury and show effects of VP in addition to the fibrotic cocktail model for snRNA Seq analysis as well as 2 IPF PLCS. Thus, taken together, we feel that the combined data is highly supportive of our findings.

R1-10. Discussion starting at 387, about the specificity of Verteporfin- would it not be preferred to inhibit Yap in both the epithelium that this paper focuses on, but also in other cell types such as the mesenchyme, that other groups have shown that Yap aberrant regulates? Is there a cell type that lung specific inhibition of Yap in this context not be adaptive?

C1-10. We agree with the reviewer that it indeed would be preferred to inhibit YAP not only in the epithelium but further in the mesenchyme, in which it has been shown to drive fibrosis. We have revised our discussion accordingly. Based on current data, while YAP signaling is involved in healthy lung homeostatic signaling, it seems that Yap activation in several lung cells within a fibrotic environment is exerting profibrotic effects. A recent

publication investigated endothelial cell phenotypes and function in persistent lung fibrosis and found a role for YAP/TAZ in sustained activation and de-differentiation of capillary endothelial cells thus contributing to persistent fibrogenesis¹. Therefore, we have added additional text in the discussion. (Line 503-505)

R1-11. Figure 2 C Acta graph is mis-spelled.

C1-11. Thank you for catching this. We have updated this and additionally checked all other figures in the manuscript and corrected minor typos.

R1-12. Line 124 should Let->led?

C1-12. Thank you. This has been corrected.

R1-13. line 266 eproduce-> produce

C1-13. Thank you. This has been corrected.

R1-14. Were the Verteporfin experiments protected from light? As previous reports have shown it to be effected by exposure.

C1-14. Thank you for this question. Yes, we took extra care to ensure that all experiments were done in the dark and the verteporfin was not exposed to light. We added text to the methods section in the revised manuscript.

R1-15. line 322, this paper also showed Yap activation in AT2 cells.

C1-15. There is indeed somewhat conflicting reports in the literature on this. In most papers, under homeostatic conditions, high amounts of nuclear Yap is not observed in AT2 cells in adult lungs. This largely matches our data whereby we do not see nuclear Yap in the majority of normal AT2 cells. However, upon injury, we see clear upregulation of nuclear Yap/Taz in AT2 cells (marked by DC-LAMP). This is now shown in murine Figure 1c, 2b and human Figure 3h and S5).

Reviewer #2 (Remarks to the Author):

R2-1. The manuscript titled "Inhibition of epithelial cell YAP-TEAD/LOX attenuates pulmonary fibrosis" by Wagner et al presents findings on the role of YAP-TEAD in pulmonary fibrosis. The study demonstrates that YAP activity leads to increased expression of LOX in fibrotic AT2 cells. Pharmacological YAP inhibition or YAP siRNA reduced LOX expression, and LOX inhibition reduced lung fibrosis in mice and in human fibrotic tissue. In conclusion, the authors highlight the potential therapy for targeting YAP-TEAD/LOX pathway for treating IPF patients.

C2-1. We thank the reviewer for the positive comments about our study and specifically for the constructive and thoughtful comments, which have been very valuable. We have

addressed the questions and comments in the revised version of our manuscript and have performed a significant number of additional experiments to further support and corroborate our findings. These new data include additional immunofluorescence staining's of the YAP/LOX axis in AT2 cells in mouse models (new Fig. 1C, 2B, and 4C) and human IPF (new Fig. 4C), additional chemical analysis using Fourier Transform Infrared Spectroscopy (FTIR) using a diamond attenuated total reflectance (ATR) accessory⁹ on lung tissue sections confirming a role for VP's inhibition of collagen crosslinking via LOX (Fig. 5F-H). We further significantly revised the manuscript text in the discussion to incorporate the discussion points raised below.

R2-2. However, while this study provides intriguing insights into the regulation of LOX expression by YAP, the impact of YAP loss in AT2 cells on reducing lung fibrosis remains controversial. Previous studies have reported conflicting results, with some showing that loss of YAP leads to reduced differentiation of AT2 cells into AT1 cells and increased lung fibrosis, while others have observed the opposite effect. The underlying reasons for this discrepancy are still unknown, making it unclear whether YAP loss-of-function is ultimately beneficial or detrimental in the context of lung fibrosis. signaling has been implicated in the regulation of MMP expression in various contexts, including cancer progression and tissue.

C2-2. We thank the reviewer for highlighting the relevance and impact of our findings describing a novel role for YAP in regulating LOX expression in AT2 cells and thus contributing to lung fibrosis. We fully agree with this reviewer that the role of YAP in AT2 cells has been controversial, which was one of our rationales to perform this study. Our study builds and extends upon recent studies that highlight the complex and dynamic roles that the Hippo-Yap/Taz pathway plays in lung development, homeostasis, and injury. We have added significant new text to the revised manuscript's discussion to highlight the cell and context dependency of the Yap and Taz in lung injury and regeneration. (Line 430-447)

We believe that our study alongside other recent publications^{3,4} help to further shed light into the role of YAP in AT2 cells specifically in a fibrotic environment. One important aspect is the potential separate roles of YAP versus TAZ, which we discuss in our manuscript (Line 510-521). We recently reported that genetic loss-of-function of YAP, but not TAZ, in AT2 cells (driven by SPC) lead to attenuated pulmonary fibrosis in vivo⁴, thus corroborating our current findings. Of note, a recent study using combined YAP/TAZ deletion in AT2 cells (in a preventive but not therapeutic approach) demonstrated increased inflammation and failed repair in two different models of alveolar injury. Recent investigations report that YAP and TAZ have independent roles as co-transcription factors and our previous work showed that YAP deletion activates TAZ signaling in the mouse model, thereby contributing to repair in vivo. Our recent work³ shows that TAZ, but not YAP, drives AT2 to AT1 regeneration while YAP induces bronchiolization in cooperation with Myc.

While VP can inhibit both YAP-TEAD as well as TAZ-TEAD interactions, it has been primarily reported to act on YAP interactions, based on affinity and protein expression levels in a given

cell. Notably, previous data and our scRNA-Seq as well as snRNA-Seq show that AT2 cells and basaloid cells have a high abundance of YAP expression, with fewer cells expressing TAZ and at lower levels. While we still need to further understand this potential mechanism, we believe that our manuscript provides further evidence of a profibrotic mechanisms driven by YAP in AT2 cells and using therapeutic approach in mouse models and ex vivo human tissues further provide evidence as a target for anti-fibrotic therapeutic intervention. We have carefully revised our manuscript text to ensure proper interpretation and discussion of other relevant studies in the field.

As part of an ongoing project intended to follow-up on the findings we present here (Stegmayr et al. *in preparation*), we have performed bulk RNA-seq on PCLS generated from regions of lung tissue with low to moderate evidence of fibrosis from patients with IPF undergoing transplant. We have treated these PCLS with multiple compounds, including verteporfin. In acknowledgment of the reviewers' concern, we show a snapshot of this forthcoming data here for reviewers only (n=4 patients). We observe decreases in MMPs associated with cancer progression (e.g. MMP7, 9 and 14) (Confidential Reviewer Figure 3), which we also observed in our snRNA-seq (new Figure 7F,G and S10). In addition, we see elevation of tumor suppressors with VP treatment such as KLF6¹⁰ and sustained or elevated SFTA2, a marker of AT2 cells, which has been shown to be positively correlated with prognosis in NSCLC patients (i.e. elevated levels of SFTA2 are associated with favorable prognosis)¹¹ (Confidential Reviewer Figure 4). Thus, while the point is well taken from the reviewer, we do not see evidence of MMP elevation, proliferation, or loss of tumor suppressors upon VP treatment.

R2-3. YAP-TEAD can directly bind to the promoters of MMP genes and regulate their transcription. Thus, it is unclear how YAP-TEAD signaling balances the signals between pro- and anti-ECM stability and maturation.

C2-3. We thank the reviewer for raising this important point. We have now re-analyzed our snRNA-seq data and extracted the most abundantly expressed MMPs in epithelial cells in IPF (new Figure S10). In addition, we have performed additional analyses on our snRNAseq data with a goal of better understanding the balance between ECM deposition and degradation. We subclustered the aberrant basaloid cells into two clusters – ABL1 and ABL2. We find that the ABL1 cluster has elevated levels and enrichment of pro-ECM related genes (e.g. collagens, fibronectin, etc) (new Figure 6C). Following treatment with VP, we found that there is a significant shift to a decrease in genes associated with fibrosis in this subcluster (Figure 6F,G), including MMP7 and MMP9. We thank the reviewer for this thoughtful comment which we feel has significantly improved our manuscript and provides further opportunities for follow-up in detail in future work.

R2-4. Lastly, AT2 cells are known as alveolar stem or progenitor cells. Their activity, including proliferation and differentiation, is critical for lung repair. It is unclear what effects LOX inhibition has on AT2 cell activity as well as senescence.

C2-4. We agree with the reviewer that targeting AT2 cells as a therapeutic target in lung fibrosis needs to consider its central role as progenitor cells. These fibrotic AT2 cells exhibit impaired progenitor cell function and display features of senescence. Several recent approaches targeting these cells led to attenuated fibrosis and repair. Most likely, this is mediated by either reversing the AT2 cell phenotype and regaining proper At2 cell function, or by blocking fibrotic signaling, remaining healthy AT2 cells can further expand and differentiate in AT2 cells. ((During the revision period, we have performed siLox experiments in At 2 cells. While we observed significant Lox knockdown, we did not see major changes in cell numbers or the expression of AT2 cell markers such as surfactant protein C. We also did not observe changes in senescence markers in our healthy or fibrotic AT2 cells upon epithelial cell knockdown of LOX. (Reviewer Figure 5))

Reviewer 5. Confirmation of siLox in normal and healthy primary fibrotic AT2 cells showing significant knockdown but no changes in senescence or the AT2 marker Sftpc.

R2-5. While this study may provide interesting findings on the role of YAP in AT2 cells in influencing ECM, the lack of scientific rationale and comprehensive characterization of YAP-TEAD activity in ECM remodeling in the setting of lung fibrosis reduces the enthusiasm toward to this study.

C2-5. We thank the reviewer for highlighting our interesting findings in the role of YAP in AT2 cells on ECM, which we believe is a novel observation in lung fibrosis with potential translational and clinical relevance. We agree that our studies open avenues to further

dissect other aspects of ECM signaling and hope to further investigate these in further studies. We hope that our additional experiments and detailed response to the reviewer comments above, address the reviewers' concerns and clarify our scientific rationale and corroborate our findings. In particular, as this reviewer and reviewer 3 asked for additional data to support ECM remodeling, we have now performed unbiased chemical and molecular analysis of healthy and fibrotic murine lung tissue from one of our two cohorts where we injected mice with verteporfin or with DMSO as a control following bleomycin administration. Excitingly, we found dramatic changes in Amide I and II which are peaks known to be associated with mature, crosslinked collagen specifically attributed to LOX. As expected, these crosslinks were elevated in lung tissue of animals with established fibrosis. However, there was a noticeable decrease in these crosslinks in animals which received verteporfin. This serves as a further, independent validation of the collagen data which we presented in our initial submission. Thus, taken together with the reduction in collagen which we observe in tissue sections, we feel that our data strongly points to a role of YAP-TEAD in regulating ECM deposition in the setting of lung fibrosis. This new data is included in Figure 5F-H.

We have also sought to add clarifying text throughout the results section to improve the rationale for each portion of our study and have further included text in the discussion regarding the importance of looking at other YAP-TEAD effects in ECM remodeling in the future. Line 503-506

Reviewer #3 (Remarks to the Author):

R3-1. In this study Wagner et al. delve into the analysis of the in vivo role of YAP signaling in lung fibrosis, focusing on the role of AT2 cells regulating extracellular matrix (ECM) remodeling. Further, the authors addressed the potential usefulness of targeting the axis YAP-TEAD/LOX as a therapeutic approach for idiopathic pulmonary fibrosis (IPF). They find that YAP signaling is early activated in fibrotic AT2 in lung fibrosis, and that pharmacological YAP inhibition by verteporfin reverses fibrotic AT2 cell reprogramming and LOX expression in experimental lung fibrosis, in vivo, and in human fibrotic tissue ex vivo. They conclude that YAP-TEAD/LOX inhibition in AT2 cells is a promising new therapy for IPF patients. The study is interesting, but it should be taken into account that the YAP dependent regulation of LOX as well as the benefit of verteporfin on fibrotic diseases have been previously reported.

C3-1. We thank the reviewer for the constructive comments and excellent suggestions. We have now addressed all questions and comments in the revised version of our manuscript, which we believe improved significantly in response to the reviewer comments. In short, we have performed a significant number of additional experiments to further support and corroborate our findings. These new data include additional immunofluorescence staining's

of the YAP/LOX axis in AT2 cells in mouse models (new Fig. 1C, 2B and 4C) and human IPF (new Fig. 3H), additional functional data using primary AT2 cells confirming a role for LOX as a crosslinking enzyme (Fig. 5). We further significantly revised the manuscript text, in particular the results to make our rationale more clear and discussion to put our study into the broader context of the most recent Yap/Taz data (Line 510-521).

R3-2. The regulatory feedback loop between YAP signaling, LOX isoenzymes and ECM remodeling has been previously described in other pathological scenarios such as pulmonary hypertension (Bertero T. et al. Cell Rep. 2015;13:1016-32) or liver fibrosis (Cheng F. et al. J Nanobiotechnology. 2023;21(1):195). These or other references on this issue should be included in the manuscript, and some categorical statements such as “... this newly identified YAP-TEAD/LOX axis” should be tempered. Authors should clearly state that the novelty of their findings specifically refers to AT2 cells.

C3-2. We thank the reviewer for highlighting these publications and have revised our manuscript additional to clarify this. We have removed the phrase ‘newly identified’ and included the references recommended by the reviewer. We would like to highlight that these papers all focus on mesenchymal Yap/Taz signaling and thus our study provides a unique perspective for the involvement of a YAP-TEAD-LOX axis in distal epithelial cells which has thus far not been reported. We hope this is more adequately outlined in the revised version.

R3-3. Introduction (last paragraph), Results and Discussion. Authors state that YAP regulates LOX expression and secretion. Because the secretion of LOX does not appear to have been directly analyzed, authors should describe their findings more properly.

C3-3. We thank the reviewer for this comment and opportunity to clarify our data. We demonstrate increased LOX secretion in multiple locations within our study and have attempted to make this more clear in the revised version. Figure 3F and 4E shows immunoblots for LOX from AT2 cell supernatants (not pellets). This definitively shows that LOX is secreted from AT2 cells and that this is regulated by YAP/TAZ (direct KD in Figure 3F and VP treatment in 4E). We also unfortunately had a typo in the figure legend of Figure 7 for Figure C. Due to the low number of samples, we did not collect tissue homogenates except for RNA. Therefore, WB was performed on concentrated PCLS supernatants which is collected from a standardized tissue volume. We have added clarifying text in the revised manuscript.

R3-4. Figure 2 C. LOX and other LOXLs should be analyzed. The authors assume that LOX is the main isoenzyme involved in IPF but there are conflicting data in the literature. Which is the contribution of each isoenzyme to the global LOX activity?

C3-4. We thank the reviewer for raising the important point of different LOX isoenzymes and apologize if this was not adequately mentioned in our initial submission. We chose to focus on LOX due to the results of our unbiased screen among the LOX family members in AT2s (Figure 3D). In our transcriptome analysis following KD of Yap/Taz in ATII cells, we found that only *Lox* was significantly downregulated by siRNA treatment with Yap/Taz. This was our initial rationale for focusing on LOX and not the other family members. Indeed, several other papers have explored the entire LOX family in IPF and shown cell type differences amongst the LOX family (Aumiller *Sci Rep* 2017 and Tjin et al. *Dis Models and Mechanisms* 2017). However, direct comparison is challenging as the amount of an enzyme or specificity of an antibody can be variable.

During the revision period, two spatial transcriptomic datasets became publicly available which we have now reanalyzed. The use of spatial transcriptomics is important as single cell workflows have been shown to bias towards certain cell populations and may thus confound results for cell type analysis. We focused on datasets obtained using the second generation of spatial transcriptomics which has been optimized for formalin fixed tissue. We performed comparable analysis as to that reported in Mayr et al. *Sci Adv* 2024 and found that transcriptionally, LOX demonstrated the most robust changes across multiple cell types in IPF samples, including in the regions enriched for fibroblasts and macrophages, but importantly also in regions enriched for alveolar cells (Reviewer Figure 5). In the revised manuscript, we have provided more detail on the subtypes of epithelial cells as this is the focus of our current work. Furthermore, we have added data for other LOX family members from our snRNAseq dataset (Figure 6H) which shows that of the LOX-family members, LOX and LOXL2 are altered in our ex vivo PCLS model and that both LOX and LOXL2 are downregulated in aberrant basaloid cells with VP treatment.

Reviewer Figure 6. Spatial transcriptomic analysis of LOX-family members in Mayr et al. *Sci Adv* 2024 with cell annotation for spot enrichment.

R3-5. In any of the studies/samples used, and in particular in human samples, did the authors observe colocalization of LOX with YAP?

C3-5. As requested by reviewer 1 also, we have now performed additional immunostainings to colocalize LOX and YAP with our mouse and human samples (Figure 3H, Figure S5 and 4C). Indeed, we observed both colocalization of YAP, HTII280 (DC-LAMP for mouse) and LOX as well as increased LOX secretion in the lumens of honeycombing regions with active, nuclear YAP.

R3-6. Figure 2F. Crosslinked collagen should be analyzed using specific techniques (e.g. LC-MS/MS or second-harmonic generation microscopy).

C3-6. We thank the reviewer for this comment. Indeed, while we measured total acid extractable collagen in Figure 2F (acid extraction is the primary mechanism for HPLC detection which we utilized), the reviewer is correct that this does not readily extract heavily crosslinked collagen. Therefore, we have now performed Fourier Transform Infrared Spectroscopy (FTIR) analysis of lung tissue sections from our mouse cohorts during the revision period. FTIR spectral peaks have been assigned to crosslinked collagen in previous studies (see Supplemental Table 3) and validated to correlate with these crosslinks as identified with LC-MS/MS specific detection of LOX crosslinks¹². We found elevated Amide I and Amide II bands in bleomycin treated animals in comparison to PBS treated animals. Encouragingly, we found changes in both of these bands upon VP treatment which correlates well with the decreased amounts of collagen we observed by HPLC as well as the reduction in modified Ashcroft score which reflects deposited collagen. As FTIR is an unbiased approach which collects the complexity of all molecules present in the tissue, we feel that this additional data strongly indicates that verteporfin treatment alters collagen crosslinking in the tissue. This data is now included in Figure 5F-H.

Furthermore, we have modified Figure 5E to make it more apparent that we have previously analyzed collagen crosslinks (beta-band). As these experiments were performed using BALF with or without BAPN and under reducing conditions with beta-mercaptoethanol (i.e. to reduce sulfide bonds but not aldehyde based crosslinking which is the mechanism by which LOX mediates crosslinking), this data strongly indicates that the fibrotic alveolar milieu is capable of regulating collagen crosslinking, and that this is largely mediated by secreted LOX (due to the dramatic reduction in the beta-band following BAPN treatment during incubation of collagen monomers with BALF (Figure 5E).

R3-7. Figure 3. LOXs should be analyzed in panel D. Immunoblot data shown in Figure 3E are not convincing due to the low number of samples. Further, LOX activity should be assessed to confirm the impact of siYAP.

C3-7. We apologize that this was not clear in the initial submission. Figure 3E in the revised manuscript (previously 3D) is simply a verification of our transcriptomics data shown in Figure 3D (previously Figure 3C). Our unbiased transcriptome analysis of fibrotic AT2 cells treated with siRNA against Yap/Taz allowed us to analyze the other Lox family members (*Lox1, 2, and 3*) as shown in Figure 3D. While we observed decreases in some samples, none reached statistical significance.

There are at present no well-established tools that can preferentially target LOX versus other LOXL family members. Indeed, BAPN has been shown to have activity against the other family members and one of the most used kits to assess LOX activity also cross-reacts with other family members. Therefore, transcriptional based analysis is, at present, the most straightforward way to assess changes in individual LOX family members.

During the revision period, we attempted to analyze LOX activity using the LOX activity kit from abcam in the secretome from supernatants from our remaining siYT treated AT2 cells. Although it is not specific for LOX (and has been used by others to detect LOXL2), we were unable to optimize the assay with the remaining samples. This may be due to the fact that our samples were lyophilized and reconstituted for use in the gelation assay.

R3-8. Figure 4. Data corresponding to Figure 4B and 4E should be confirmed at the level of LOX activity or collagen crosslinking.

C3-8. Thank you for pointing out that we have written this in a confusing manner. As mentioned above, due to the lack of tools available with which to study LOX activity, we have chosen to assess its activity through the development of the novel collagen gelation assay we described here. As LOX (and other family members) play a role in intermolecular crosslinking (i.e. crosslinking neighboring collagen monomers), we hypothesized that the presence of LOX would alter collagen assembly. We therefore used bronchioalveolar lavage fluid from our *in vivo* murine cohort. Furthermore, we aimed to prove that the disease-specific changes in collagen deposition we observed between healthy and fibrotic BALF were in part due to the presence of active LOX. For this, we used BAPN which is the most widely used pharmacological tool to assess or manipulate LOX activity^{13,14}. Because we have used the same amount of starting collagen type I in all assays, differences in the **speed** at which the gel forms and the **maximum gelation** are driven by accessory molecules such as LOX but also can be driven by proteoglycans. Our data indicates that the speed at which gelation occurs is not affected by BAPN, but that the overall maximum gelation (a reflection of crosslinking in the network) is affected. This matches well with data showing that proteoglycans present during collagen assembly can affect the speed at which gelation occurs (such as versican (Chen et al. Sci Reports 2020, doi: 0.1038/s41598-020-76107-0)). Therefore, we feel that this data strongly indicates that LOX is responsible for these changes.

As LOX induces specific cross links, we have now undertaken additional experiments during the revision period which addresses this point on a biochemical level. We have performed Fourier Transform Infrared Spectroscopy (FTIR) on our *in vivo* tissue sections in ATR-Transmission mode which allows us to analyze the molecular composition (including protein secondary structures) in an unbiased way. We collected spectra for the entire range capable with our FTIR (Agilent, Cary630) (i.e. 400-4000cm⁻¹). As expected, we detected multiple Amide bands in regions well-documented in the literature (i.e. Amide A and B at 3300 and 3000cm⁻¹ respectively as well as Amide I and II at 1650 and 1550cm⁻¹). (See Supplemental Table 3) Next, we sought to identify the contribution of these two bands as the 1650 and 1550 band. Interestingly, a portion of the Amide I band has recently been shown to be specifically due to LOX and to correlate well with the amount of LOX and pyridinoline crosslinks assessed in the same study using LC-MS¹². Attribution of the 1660 cm subband to LOX has been shown in multiple independent papers¹⁵, including in experiments which selectively removed these crosslinks¹². This new data is in Figure F-H.

R3-9. Figure 5. Data from experiments using BAPN, should be confirmed in LOX knockdown experiments. As indicated above a more reliable method to assess crosslinked collagen should be used (panel E).

C3-9. During the revision period, we performed siLox on normal and fibrotic AT2 cells which was confirmed on the transcript level (see Reviewer Figure 5). Furthermore, we had previously utilized samples from siYT and VP treated normal and fibrotic AT2 cells. Unfortunately, as before, we were unable to detect any differences in our collagen gelation assay between normal and fibrotic AT2 cells. Both fibronectin¹⁶ and vitronectin¹⁷ have been shown to affect the speed and maximum gelation in collagen turbidity assays with both accelerating the rate at which collagen is assembled and furthermore increasing maximum gelation. As our AT2 cells are cultured with FBS and both are known components of FBS, it is presently not possible to utilize the turbidity assay for assessment of collagen assembly due to this confounding factor. As evidence of this, we have included additional controls in a final experiment where we have used nearly 1.5 times the amount of collagen from our starting amount (1 mg/mL versus 1.45mg/mL) to show that even increasing the collagen amount does not impact the collagen assay with regard to speed but minimally does for maximum gelation. This demonstrates the disproportionate role that accessory molecules have during collagen assembly.

siLox

Reviewer Figure 7. Collagen gelation assay during the first two hours for siYT, VP and siLOX treatment of normal and fibrotic (from top to bottom)

R3-10. In many studies the number of replicates per experiment is low: in many cases only 3 per group, and in immunoblots only 2 or 1 lanes/samples per experimental condition are shown.

C3-10. We acknowledge this point from the reviewer. Due to the time-consuming nature of some of these experiments (and the rarity of IPF lung transplants which can be sliced), we were unable to add additional n's during the study period. We therefore show these as representative data points, do not apply statistics, and ensure proper conclusion in the text.

In our response to Reviewer 2, point 2, we have shown additional data which is planned for subsequent manuscripts. In addition to the questions raised by Reviewer 2, we have also extracted the key genes shown in Figure 7 of the current manuscript where we show an n=2 for IPF patients (Confidential Reviewer Figure 8). As mentioned above, the current manuscript will be followed up by a subsequent manuscript which further analyses the effects of VP in comparison to other compounds in IPF tissue from regions with only little to no visible fibrotic changes.

To address the fact that we showed representative images for Figure 5B, we have performed detailed image analysis on the SEM images using TWOMBLI which was developed for complex image analysis of diverse extracellular matrix imaging datasets¹⁸. We have analyzed all the SEM data that we have from individual experiments spanning 3 independent primary AT2 isolations. While not originally designed or validated for SEM images, we found that we were successfully able to implement this tool with some modifications in brightness/contrast with our images. We extracted values for lacunarity and fractal dimensions as we deemed these most relevant for our dataset and they have been applied specifically to collagen fibers¹⁹. Lacunarity is a measure of how the ECM fills the space (i.e. the number and size of gaps in the matrix). Larger values indicate larger space in the matrix pattern. On the other hand, fractal dimensions have been shown to be useful in analysis of collagen in diverse applications, including SEM images of tendons²⁰. While we did not reach statistical significance due to the fact that the power calculation for these experiments was done with transcriptomic data as the intended endpoint, we find clear trends which support our other findings that the YAP-TEAD axis in fibrotic AT2 cells alters ECM arrangement. This new analysis is included in supplement figure 6.

R3-11. Supplementary Figure S1 only consists of one panel, therefore S1A does not exist.

C3-11. Thank you for catching this. We have corrected this in the main text.

R3-12. Figure 7A is referred in the text before Figure 6A. Panel 7A should be included in Figure 6 for easy reading.

C3-12. Thank you for catching this mistake. Figure 7A was inadvertently mentioned in the text. It appropriately refers to the experiments in Figure 7 and not those in Figure 6 which were an independent cohort intended for snRNA-seq. We have removed this from the manuscript and further modified Figure 7 to more clearly show the two different experimental setups.

R3-13. All tables should be referred as “supplementary table” and numbered according to its order of appearance in the text.

C3-13. Thank you for pointing this out. We have revised all tables.

R3-14. The sentence in line 292 is speculative since ECM stiffness was not measured.

C3-14. Thank you. We have revised this sentence.

R3-15. In the immunoblot shown in Figure 7C a loading control should be included.

We have now added the loading control for these experiments. As these are from supernatants of 4mm biopsy punched PCLS, all samples are normalized to the same volume of supernatant. Nonetheless, we used stain-free gel technology which labels tryptophans and can further serve as a loading control. We have included the same 3 patients in Figure 7C as the original submission. Due to the aberrant spreading of bands present at low molecular weights in Patient 1 and 2 under FC conditions, we have also included the other two patients used for quantification in Figure S9 to show that this is not present in all samples.

R3-16. Discussion. The assumption that LOX is not induced by TGFbeta (lines 365-366) is not correct. This issue is known from 90's (please see the review by Laczko and Csiszar Biomolecules 2020).

C3-16. We thank the reviewer for this comment and references to this review paper about LOX function. We have included this reference and further revised our manuscript text accordingly, which now reads as follows.

LOX has been shown to be regulated by TGF- β 1 in several fibrotic conditions²¹ and additionally has been shown to be capable of regulating TGF-beta signaling in experimental pulmonary fibrosis²². However, this appears to be cell type and context dependent. Previous work has shown, for example, that LOXL2, but not LOX expression was induced by TGF- β 1 in lung epithelial cells⁷. (Line 464-466) Similarly, LOX but not LOXL2 was identified as a downstream target of YAP-TEAD driven transcription in breast epithelial cells in one of the seminal studies that identified TEADs as the main transcription factor family interacting with YAP identified⁸. A clinical trial targeting LOXL2 using the monoclonal antibody simtuzumab, found no benefit regarding survival²³, although limited data about its bioavailability and target engagement in IPF are known. In line with this, our data indicate that YAP-TEAD driven LOX expression in distal lung epithelial cells might be driven by other upstream factors, including Hippo signaling components. Future work should more specifically look at crosstalk with TGF- β signaling. (Line 535-537)

R3-17. Please check spelling through the manuscript (e.g. limted, interstingly,)

C3-17. Thank you for highlighting this. We have done our best to catch all typos throughout.

References

- 1 Raslan, A. A. *et al.* Lung injury-induced activated endothelial cell states persist in aging-associated progressive fibrosis. *Nature Communications* **15**, 5449, doi:10.1038/s41467-024-49545-x (2024).
- 2 Mayr, C. H. *et al.* Spatial transcriptomic characterization of pathologic niches in IPF. *Science Advances* **10**, ead15473, doi:10.1126/sciadv.adl5473.
- 3 Warren, R. *et al.* Cell competition drives bronchiolization and pulmonary fibrosis. *Nature Communications* **15**, 10624, doi:10.1038/s41467-024-54997-2 (2024).
- 4 Warren, R., Lyu, H., Klinkhammer, K. & De Langhe, S. P. Hippo signaling impairs alveolar epithelial regeneration in pulmonary fibrosis. *Elife* **12**, doi:10.7554/eLife.85092 (2023).
- 5 Shiraishi, K. *et al.* Biophysical forces mediated by respiration maintain lung alveolar epithelial cell fate. *Cell* **186**, 1478-1492.e1415, doi:<https://doi.org/10.1016/j.cell.2023.02.010> (2023).
- 6 Burgess, C. L. *et al.* Generation of human alveolar epithelial type I cells from pluripotent stem cells. *Cell Stem Cell* **31**, 657-675.e658, doi:10.1016/j.stem.2024.03.017 (2024).
- 7 Aumiller, V. *et al.* Comparative analysis of lysyl oxidase (like) family members in pulmonary fibrosis. *Sci Rep* **7**, 149, doi:10.1038/s41598-017-00270-0 (2017).
- 8 Zhao, B. *et al.* TEAD mediates YAP-dependent gene induction and growth control. *Genes Dev* **22**, 1962-1971, doi:10.1101/gad.1664408 (2008).
- 9 de Campos Vidal, B. & Mello, M. L. S. Collagen type I amide I band infrared spectroscopy. *Micron* **42**, 283-289, doi:<https://doi.org/10.1016/j.micron.2010.09.010> (2011).
- 10 Ito, G. *et al.* Krüppel-like factor 6 is frequently down-regulated and induces apoptosis in non-small cell lung cancer cells. *Cancer Res* **64**, 3838-3843, doi:10.1158/0008-5472.Can-04-0185 (2004).
- 11 Li, N., Zhai, Z., Chen, Y. & Li, X. Transcriptomic and immunologic implications of the epithelial-mesenchymal transition model reveal a novel role of SFTA2 in prognosis of non-small-cell lung carcinoma. *Front Genet* **13**, 911801, doi:10.3389/fgene.2022.911801 (2022).
- 12 Mieczkowska, A. & Mabileau, G. Validation of Fourier Transform Infrared Microspectroscopy for the Evaluation of Enzymatic Cross-Linking of Bone Collagen. *Calcified Tissue International* **113**, 344-353, doi:10.1007/s00223-023-01105-z (2023).
- 13 Jones, M. G. *et al.* Nanoscale dysregulation of collagen structure-function disrupts mechano-homeostasis and mediates pulmonary fibrosis. *eLife* **7**, e36354, doi:10.7554/eLife.36354 (2018).
- 14 Tjin, G. *et al.* Lysyl oxidases regulate fibrillar collagen remodelling in idiopathic pulmonary fibrosis. *Dis Model Mech* **10**, 1301-1312, doi:10.1242/dmm.030114 (2017).
- 15 Paschalis, E. P. *et al.* Spectroscopic Characterization of Collagen Cross-Links in Bone. *Journal of Bone and Mineral Research* **16**, 1821-1828, doi:<https://doi.org/10.1359/jbmr.2001.16.10.1821> (2001).
- 16 Paten, J. A. *et al.* Molecular Interactions between Collagen and Fibronectin: A Reciprocal Relationship that Regulates De Novo Fibrillogenesis. *Chem* **5**, 2126-2145, doi:<https://doi.org/10.1016/j.chempr.2019.05.011> (2019).

- 17 Date, K., Sakagami, H. & Yura, K. Regulatory properties of vitronectin and its glycosylation in collagen fibril formation and collagen-degrading enzyme cathepsin K activity. *Scientific Reports* **11**, 12023, doi:10.1038/s41598-021-91353-6 (2021).
- 18 Wershof, E. *et al.* A FIJI macro for quantifying pattern in extracellular matrix. *Life Science Alliance* **4**, e202000880, doi:10.26508/lsa.202000880 (2021).
- 19 Mostaçõ-Guidolin, L. B. *et al.* Fractal dimension and directional analysis of elastic and collagen fiber arrangement in unsectioned arterial tissues affected by atherosclerosis and aging. *Journal of Applied Physiology* **126**, 638-646, doi:10.1152/jappphysiol.00497.2018 (2019).
- 20 Frisch, K. E. *et al.* Quantification of collagen organization using fractal dimensions and Fourier transforms. *Acta Histochem* **114**, 140-144, doi:10.1016/j.acthis.2011.03.010 (2012).
- 21 Laczko, R. & Csiszar, K. Lysyl Oxidase (LOX): Functional Contributions to Signaling Pathways. *Biomolecules* **10**, 1093 (2020).
- 22 Cheng, T. *et al.* Lysyl oxidase promotes bleomycin-induced lung fibrosis through modulating inflammation. *J Mol Cell Biol* **6**, 506-515, doi:10.1093/jmcb/mju039 (2014).
- 23 Raghu, G. *et al.* Efficacy of simtuzumab versus placebo in patients with idiopathic pulmonary fibrosis: a randomised, double-blind, controlled, phase 2 trial. *Lancet Respir Med* **5**, 22-32, doi:10.1016/s2213-2600(16)30421-0 (2017).

REVIEWER COMMENTS

Reviewer #1 (Remarks to the Author):

This manuscript from Wagner et al/ the Konigshoff group is well written and demonstrates that activation of Yap in IPF AT2 cells is associated with activation of Lox and subsequent fibrotic remodeling. Inhibition of this Yap/Lox axis with the Yap/Tead inhibitor Verteporfin reduces fibrotic remodeling in mouse models of fibrosis at 14 days, and reduces fibrotic signaling in human lung sections. The data strongly supports these concepts, and the revisions are well done and greatly enhance the manuscript. My concerns were thoroughly addressed.

Thank you for your previous comments and support of our work.

Reviewer #2 (Remarks to the Author):

1. The new data that are used to address reviewers comments, including “Confidential Reviewer 5”, should be provided in the revised manuscript.

We thank the reviewer for their previous questions which led to these experiments. We are happy to include the Reviewer 5 data in the revised text as it is not planned to be used in any other studies. The results are now included in Figure S8 with detailed methods given in the main text. We agree that this is important to show in order to support future strategies which aim to target this axis to demonstrate that inhibition of LOX does not induce senescence nor change the phenotype of ATII cells.

2. Figures 6 and 7 provided interesting data on the effects of VP on lung fibrosis. However, a critical question remains unanswered: which cell type is the primary target of VP’s action? Both fibroblasts and epithelial cells can produce LOX. Unfortunately, this study does not distinguish between VP’s effects on these cell types, leaving uncertainty about which cell type contributes most significantly to the observed anti-fibrotic effects.

This is an excellent question and one which we are looking forward to address in extensive work in follow-up studies. This will be important to better understand the effects of verteporfin, or subsequent drugs targeting this axis. In acknowledgment of this, we have now performed new analyses with the snRNA-seq dataset to look at effects of VP on fibroblast, endothelial and immune populations which we also detected in our dataset. We find that the YAP-TEAD target genes CCN1 and CCN2 are induced by FC treatment and subsequently reduced by VP. However, there are also moderate changes in fibroblasts with FC treatment as well. Noticeably, we do not see

robust induction of myofibroblasts in our dataset (as evident by incremental increases in ACTA2 and no changes in COL8A1, which has been described as an early and specific fibroblast marker. We have also included changes in LOX, LOXL1, and LOXL2 across all cell types. This new data can be found in S12 and S15.

3. Figure 2A showed that doses of 2 U/kg bleomycin caused about 37% mice lethality. However, based on the published research results, 2 U/kg bleomycin treatment typically does not cause mouse lethality.

We thank the reviewer for pointing out that we may not have been clear enough about the animal model that we used and our terminology regarding survival. We randomized all animals into one of four groups prior to the experiment (PBS-vehicle, PBS-VP, Bleo-vehicle, Bleo-VP). We are utilizing a model of bleomycin where we intubate the animals and administer the bleomycin intratracheally. Our weight loss curves are in line with our previous work/historic data and other literature using intratracheal bleomycin administration (<https://pmc.ncbi.nlm.nih.gov/articles/PMC11595013/#app1-ijms-25-12300> or <https://www.nature.com/articles/ncomms14532> , Figure S1B). Therefore, we have now included a new Figure S2B which shows our weight loss data over time.

According to local ethics laws in Bavaria, Germany, where the in vivo studies were conducted, animals cannot lose more than 15% body weight without euthanasia. We direct the reviewer to lines 591-592. “Animals were weighed every day to monitor their health; animals which lost more than 15% body weight from the study onset were euthanized.”

4. Figure 3H: The co-localization of LOX, HTII-280, and YAP is not clear.

We thank the reviewer for their question. We have now provided digital magnifications (2x) of two regions of interest found in 3H (provided in new supplement figure S6). We hope this helps to clarify and demonstrate that HTII-280+, YAP+ cells are LOX+ and that LOX is secreted into the lumens/apically in these regions. Our LOX antibody recognizes both the pro and secreted form but we predominantly are interested in secreted LOX due to extracellular LOX's role in collagen crosslinking. Therefore, colocalization is not our primary aim as this would be with intracellular, pro-LOX. Nonetheless, we see elevated , intracellular pro-LOX expression in YAP+ airways and we also observe extracellular LOX in close proximity to HTII-280 and YAP+ATII cells. We have included a higher magnification for this in the revised manuscript and have therefore reworded ‘co-localization’ in lines 218-219.

5. Figure 7C: It's unclear if LOX expression was really increased in FC, since the total protein loading for FC appeared to be higher than CC. This makes the quantitative interpretation unreliable.

We thank the reviewer for their question. Indeed, the reviewer is correct, in some patients we see general increases in protein secretion with the FC treatment in comparison to CC. This is also something we observed in our initial study describing the FC model (see Figure 3C in Alsafadi et al, AJP-Lung 2017). As in our previous work, we have chosen to load the same amount of concentrated supernatant volume per lane in our SDS page gels due to the fact that the supernatant in each condition is collected from the same corresponding PCLS tissue volume. Therefore, our approach is analogous to determining protein concentration in the secreted volume as opposed to protein amount in relation to total secreted proteins. We feel that showing the data in this way is important due to the fact that global increases in protein secretion per unit of tissue volume (and therefore cells) has high biological relevance, especially in fibrosis.

*To illustrate this point more clearly, we have performed normalization to the stain-free gel and include this in a figure for the Reviewers. In the western blots shown in the manuscript (Figure 7D and S14), it is visually apparent that there is more LOX secretion with FC in each patient as compared to time matched CC PCLS. Importantly, the reduction with VP in the FC condition is still statistically significant with this data presentation approach (see Rev Figure 1). However, the increased LOX secretion with FC treatment appears blunted when normalized to 'total' secreted protein. The stain-free technology is an imperfect method of protein normalization and is especially problematic in this experimental setup for quantification purposes. The stain-free technology labels tryptophan amino acids, which are completely absent in collagens. We and others (Machahua et al. Resp Res 2025) have shown that collagen secretion is upregulated with FC treatment. Therefore, the stain free can only **estimate** the amount of protein per lane that contains tryptophan. Other protein normalization assays, such as BCA, etc. can be used but collagen is notoriously difficult to accurately account for in these assays due to its unique amino acid sequence and structure. Therefore, we feel that the most technically correct way to show this data in the manuscript is to quantify the LOX bands only as they are normalized to a standard amount of PCLS supernatant volume.*

Reviewer Figure 1. Quantification of LOX secretion in PCLS at day 5 with and without FC or VP treatment. LOX is normalized to total secreted protein, as assessed by stain-free technology. Each line represents an individual patient corresponding to Figure 7D and Supplemental S14.

We want to emphasize that the reduction of LOX with VP in the FC condition is statistically significant no matter how we present the data. We are thus confident in the finding that VP reduces LOX secretion (both total amount of LOX secreted as well as specifically reduces LOX secretion in relation to other proteins).

Reviewer #3 (Remarks to the Author):

No further questions

Thank you for the previous input you have given on improving our manuscript.